# Sexually dimorphic architecture and function of a mechanosensory circuit in *C. elegans*

Hagar Setty [1,2,4], Yehuda Salzberg[1,2,4], Shadi Karimi[3], Elisheva Berent-Barzel [1,2], Michael Krieg [3] & Meital Oren-Suissa [1,2] ✉

How sensory perception is processed by the two sexes of an organism is still only partially understood. Despite some evidence for sexual dimorphism in auditory and olfactory perception, whether touch is sensed in a dimorphic manner has not been addressed. Here we find that the neuronal circuit for tail mechanosensation in *C. elegans* is wired differently in the two sexes and employs a different combination of sex-shared sensory neurons and interneurons in each sex. Reverse genetic screens uncovered cell- and sex-specific functions of the alpha-tubulin *mec-12* and the sodium channel *tmc-1* in sensory neurons, and of the glutamate receptors *nmr-1* and *glr-1* in interneurons, revealing the underlying molecular mechanisms that mediate tail mechanosensation. Moreover, we show that only in males, the sex-shared interneuron AVG is strongly activated by tail mechanical stimulation, and accordingly is crucial for their behavioral response. Importantly, sex reversal experiments demonstrate that the sexual identity of AVG determines both the behavioral output of the mechanosensory response and the molecular pathways controlling it. Our results present extensive sexual dimorphism in a mechanosensory circuit at both the cellular and molecular levels.

In sexually reproducing species, evolutionary forces have shaped the nervous systems of the two sexes such that they feature overt sex-specific behaviors. These dimorphic behaviors can result from differences in sensory neuron perception, downstream neuronal processing, or both. Sexually dimorphic sensory perception has been observed in both vertebrate and invertebrate models. In *C. elegans*, the differential expression of a G-protein-coupled receptor (GPCR) in a sensory neuron determines the different behavioral outcome[1]. In frogs, sexually dimorphic auditory tuning has been documented[2–4]. In mice, sexually dimorphic olfactory perception[5,6] and integration of sensory information were observed. The sexually dimorphic behavior in response to the pheromone ESP1, for example, was shown to be mediated through dimorphic processing in third- and fourth-order brain areas[7,8]. Integration downstream to

the sensory level is also evident in aggressive behavior in *Drosophila* and nociceptive behavior in *C. elegans*, where sexually dimorphic processing by downstream interneurons regulates the dimorphic behavior[9,10]. These examples demonstrate that for different types of sensory modalities, be it olfactory, auditory or chemo-aversion, sex differences can originate from dimorphism in distinct neuronal layers.

The perception of mechanical forces, or mechanosensation, includes the perception of touch, hearing, proprioception and pain, all of which involve the transduction of mechanical forces into a cellular signal[11,12]. Numerous studies have focused on understanding the molecular and cellular pathways involved in mechanosensation in both mammalian and invertebrate systems[13–15]. Although recent evidence has provided insight into sex differences in mechanical

[1]Department of Brain Sciences, Weizmann Institute of Science, Rehovot 7610001, Israel. [2]Department of Molecular Neuroscience, Weizmann Institute of Science, Rehovot 7610001, Israel. [3]ICFO-Institut de Ciències Fotòniques, The Barcelona Institute of Science and Technology, 08860 Castelldefels, Barcelona, Spain. [4]These authors contributed equally: Hagar Setty, Yehuda Salzberg. ✉e-mail: meital.oren@weizmann.ac.il

nociception[16–20], it is still unknown whether males and females sense innocuous touch differently.

*C. elegans* has been used extensively to explore the cells and molecules that govern mechanosensation, and specifically touch sensation[15,21]. Touch sensation in this species can generally be divided into two different modalities: gentle touch and harsh touch, each involving different cellular and molecular mechanisms[22–25]. The cellular mechanisms mediating harsh mechanical stimulation applied to the tail, or tail mechanosensation, have only recently begun to be examined. Studies focused on harsh-touch-induced tail mechanosensation discovered that the sensory neurons PHA, PHB and PHC and the interneurons PVC and DVA are required for tail mechanosensation in hermaphrodites[25,26]. These studies, however, did not identify the molecular mechanisms mediating touch sensitivity in these cells, nor did they address whether the same cells constitute or are part of the male tail mechanosensation circuit.

The neuronal connectome of both sexes in *C. elegans*[27–29] indicates that many circuits, including those mediating mechanosensation, contain sexually dimorphic connections (Fig. 1a). Here we report that while males and hermaphrodites respond similarly to harsh touch of the tail, they sense and integrate mechanosensory information differently, in terms of the neurons involved, the underlying circuits and the molecular components mediating and transducing touch. Our results establish the circuit architecture for tail mechanosensation in both sexes and reveal a role for the sex-shared interneuron AVG as an integrator of mechanosensory information. We further identify several key components of the molecular machinery controlling the activity of this circuit and reveal the sex-specific functions of these molecules at both the sensory and interneuron levels. Our results demonstrate that the propagation of harsh-touch tail mechanosensory information is sexually dimorphic and provide a unique example of how neuronal circuits evolved sex-specific features while maintaining the same sensory modality and its behavioral output.

## Results

### Sensory level sexual dimorphism in the tail mechanosensation circuit

By combining data from the published connectome maps of both *C. elegans* sexes[27] with behavioral data from previous studies[25,26], we revealed that some of the sensory cells in the tail mechanosensation circuit are connected differently in the two sexes (Fig. 1a). To explore the contribution of each sensory neuron to tail mechanosensation, we silenced individual neurons by cell-specific expression of the inhibitory *Drosophila* histamine-gated chloride channel (HisCl1)[30] and then tested both sexes for tail-touch response (see *Methods*). Silencing the PHB neuron diminished the tail-touch response equally in both sexes, suggesting it has a sex-independent role in touch sensitivity (Fig. 1b). However, silencing PHA and PHC revealed that PHC is necessary only in hermaphrodites (corroborating previous findings)[26], while PHA is required only in males (Fig. 1b). These results show that at the sensory level, each sex utilizes a different combination of sensory cells in tail-touch perception.

We next asked whether the molecular mechanisms that govern tail mechanosensation at the sensory level are, too, sexually dimorphic. We carried out a reverse genetic screen targeting ion channels and other proteins previously suggested to be involved in mechanosensation that are known to be expressed in the tail sensory cells[22,31–37] (Fig. 1c). Assaying tail-touch responses of RNA interference (RNAi)-fed or mutant animals led to the identification of two genes whose silencing caused sex-specific defects in tail mechanosensation.

First, targeting *mec-12* by using both RNAi or a *mec-12* mutant, reduced tail mechanosensation only in males (Fig. 1d, e). *mec-12* (expressed in PHA, PHB and PHC, Fig. 1c) encodes an alpha-tubulin protein, is one of several genes required for touch receptor neuron

(TRN) function in *C. elegans* and is specifically responsible for generating 15-protofilament microtubules in TRNs[38].

Second, we found that RNAi of *tmc-1* elicited a sexually dimorphic response, and *tmc-1* mutant hermaphrodites exhibited a significantly reduced tail-touch response (Fig. 1d, e). TMC-1 (expressed in PHA and PHC, Fig. 1c) is a mechanosensitive ion channel and an ortholog of the mammalian TMC proteins important for hair-cell mechanotransduction[32,39–41]. Taken together, our screen uncovered two molecules that play a role in mediating tail mechanosensation in a sex-dependent manner, possibly functioning through different types of sensory cells (Fig. 1f).

### Cell- and sex-specific function of *mec-12* and *tmc-1* in tail mechanosensation

Having established a role for *mec-12* in mechanosensation in males, we turned to explore whether it functions sex-specifically through the phasmid neurons. Since PHA and PHB are required for tail mechanosensation in males (Fig. 1b), we restored the expression of *mec-12* in mutant animals under either the *srg-13* or the *gpa-6* promoters, which drive expression in PHA and PHB respectively, or the *che-12* promoter, which drives expression in ciliated neurons, including PHA and PHB[42,43]. We found that *mec-12* expression in PHA but not PHB is sufficient to rescue the tail touch phenotype (Fig. 2a, Supplementary Fig. 1a), suggesting that *mec-12* functions in PHA. Supporting this finding, we observed the expression of the *mec-12* gene and protein in PHA in males (Supplementary Fig. 1b, c). To further examine whether *mec-12* expression is sufficient to render an animal sensitive to tail touch, we exploited the fact that *tmc-1* mutant hermaphrodites have a reduced tail-touch response (Fig. 1e), and force-expressed *mec-12* in PHA in *tmc-1* mutant hermaphrodites. Indeed, *tmc-1* mutant hermaphrodites which express *mec-12* specifically in PHA restored their tail-touch responses (Fig. 2b), strongly arguing for a functional role for *mec-12* in tail mechanosensation.

*tmc-1* was shown to be expressed in PHA and PHC in hermaphrodites[34] (Fig. 1c). Since PHC is required for tail mechanosensation in hermaphrodites and not PHA (Fig. 1b), we hypothesized that *tmc-1* mediates tail mechanosensation through PHC in hermaphrodites. Indeed, expressing *tmc-1* specifically in PHC rescued the defective tail-touch response of *tmc-1* mutant hermaphrodites (Fig. 2c). This finding is supported by the evident expression of a *tmc-1::mKate2* transcriptional reporter in PHC in both sexes (Supplementary Fig. 1d). Taken together, our results demonstrate that not only do different sensory neurons participate in the processing of mechanosensory information in each sex, the molecules involved in this processing in the sensory neurons are, too, distinct between the sexes (Fig. 2d).

### The tail mechanosensation circuit is sexually dimorphic at the interneuron level

The predicted sexually dimorphic connectivity of the sensory neurons (Fig. 1a) suggests dimorphic activities for the downstream interneurons. The sex-shared interneuron AVG is predicted to possess a striking dimorphic connectivity pattern according to the published connectomes, receiving more inputs in males compared to hermaphrodites (Supplementary Fig. 2). Given that the sensory cells connected to AVG in males are the ones with a suggested role in mechanosensation, we speculated that AVG might be involved in tail mechanosensation. Therefore, we silenced AVG using a cell-specific driver (Supplementary Fig. 3a, b) and tested animals for tail-touch responses in both sexes. We found that silencing AVG elicits a sexually dimorphic effect on tail mechanosensation, reducing only the male response (Fig. 3a), in agreement with AVG's predicted dimorphic connectivity.

If AVG connectivity to the sensory neurons plays a significant role, rewiring AVG's connections should affect tail-mechanosensation responses. Namely, adding connections in hermaphrodites would

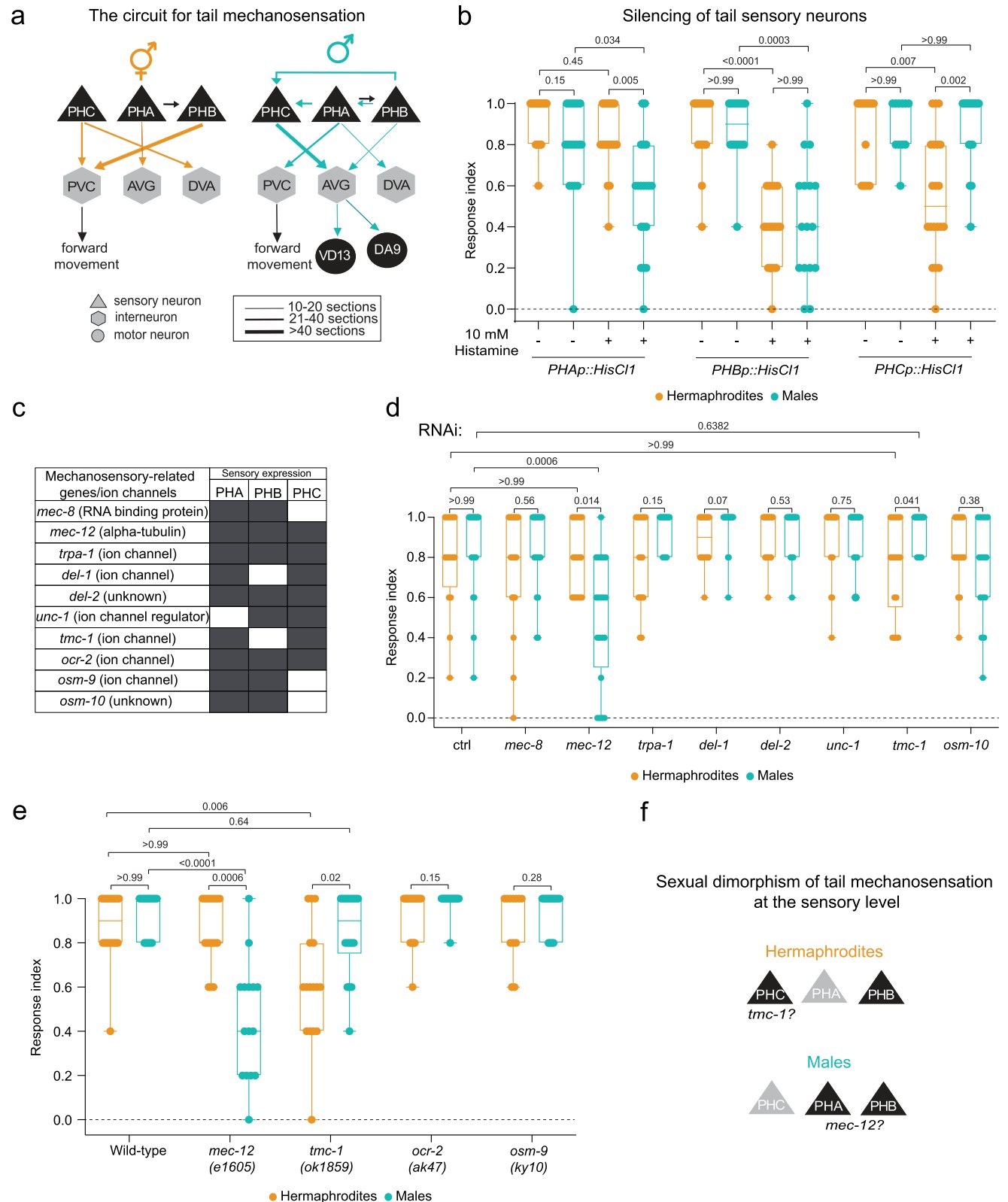

elicit an effect, and vice-versa for males. We therefore manipulated the sex-determination pathway of the worm specifically in AVG to switch its sexual identity and connectivity to that of the opposite sex[43–45]. Sex-reversals were done by manipulating the activity of the Gli transcription factor TRA-1, the master regulator of sexual differentiation in *C. elegans*[46,47]. TRA-1 is expressed ubiquitously in hermaphrodites and is thought to autonomously control cell-type-specific transcription programs that, in turn, control specific sexually dimorphic features[48]. In

males, TRA-1 is downregulated via protein degradation[49,50]. To feminize, we expressed the intracellular domain of TRA-2, tra-2(ic), which constitutively signals to prevent TRA-1 downregulation and to masculinize, we cell-specifically expressed FEM-3[44,45]. We found that sex-reversing AVG is sufficient to convert the phenotype of the AVG-silenced tail-touch response: In hermaphrodites with a masculinized AVG, the tail-touch response was impaired when AVG was silenced compared to wild-type hermaphrodites, and in males with feminized

**Fig. 1 | Sexually dimorphic perception of tail mechanosensation at the sensory level. a** Predicted connectivity of the circuit for tail mechanosensation[27,29]. Chemical synapses between sensory (triangles), inter- (hexagons) and motor (circles) neurons are depicted as arrows. Thickness of arrows correlates with degree of connectivity (number of sections over which *en passant* synapses are observed). **b** Tail-touch responses of *PHAp::HisCl1-*, *PHBp::HisCl1-* and *PHCp::HisCl1-*expressing animals of both sexes that were tested either on histamine or control plates (see *Methods*). Number of animals: *PHAp::HisCl1* hermaphrodites: $n = 19$ per group, *PHAp::HisCl1* control males: $n = 18$, *PHAp::HisCl1* histamine males: $n = 19$, *PHBp::HisCl1* control hermaphrodites: $n = 16$, *PHBp::HisCl1* histamine hermaphrodites: $n = 18$, *PHBp::HisCl1* males: $n = 16$ per group, *PHCp::HisCl1* control hermaphrodites: $n = 15$, *PHCp::HisCl1* control males: $n = 12$, *PHCp::HisCl1* histamine hermaphrodites: $n = 20$, *PHCp::HisCl1* histamine males: $n = 16$. **c** Table listing the genes selected for reverse genetic screen and their expression pattern at the sensory neurons PHA, PHB and PHC. Dark boxes represent gene expression. **d** Tail-touch responses of RNAi-silenced candidate genes in both sexes. Control hermaphrodites: $n = 16$, control males: $n = 19$, *mec-8* RNAi: $n = 19$ animals per group,

*mec-12* RNAi hermaphrodites: $n = 17$, *mec-12* RNAi males: $n = 20$, *trpa-1* RNAi: $n = 16$ per group, *del-1* RNAi hermaphrodites: $n = 18$, *del-1* RNAi males: $n = 17$, *del-2* RNAi hermaphrodites: $n = 19$, *del-2* RNAi males: $n = 17$, *unc-1* RNAi hermaphrodites: $n = 19$, *unc-1* RNAi males: $n = 20$, *tmc-1* RNAi: $n = 18$ per group, *osm-10* RNAi hermaphrodites: $n = 16$, *osm-10* RNAi males: $n = 15$. **e** Tail-touch responses of mutant strains for candidate genes in both sexes and in *him-5(e1490)* animals (see *Methods*). *him-5(e1490)* hermaphrodites: $n = 18$, *him-5(e1490)* males: $n = 15$, *mec-12(e1605)* hermaphrodites: $n = 17$, *mec-12(e1605)* males: $n = 15$, *tmc-1(ok1859)* hermaphrodites: $n = 16$, *tmc-1(ok1859)* males: $n = 18$, *ocr-2(ak47)* hermaphrodites: $n = 16$, *ocr-2(ak47)* males: $n = 15$, *osm-9(ky10)* hermaphrodites: $n = 12$, *osm-9(ky10)* males: $n = 15$. **f** Schematic of the sensory neurons which function in each sex, with the respective suggested sites of action of *tmc-1* and *mec-12*. The response index represents an average of the forward responses (scored as responded or not responded) in five assays for each animal. In (**b**, **d**, **e**) we performed a Kruskal-Wallis test followed by Dunn's multiple comparison test. Orange- hermaphrodites, cyan- males. All Bar graphs are a box-and-whiskers type of graph, min to max showing all points. The vertical bars represent the median.

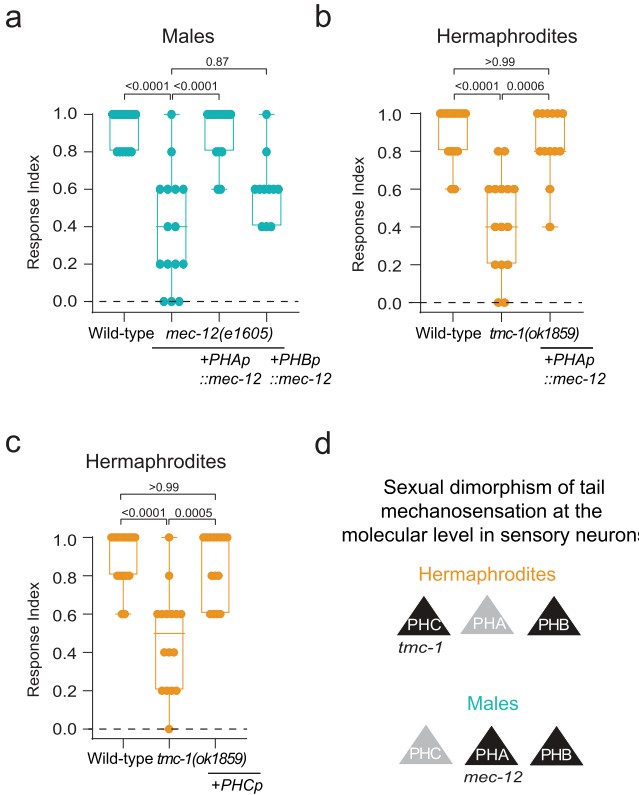

**Fig. 2 | *mec-12* and *tmc-1* function cell- and sex-specifically in tail mechanosensation. a** Tail-touch responses of wild-type ($n = 15$), *mec-12(e1605)* ($n = 15$), *mec-12(e1605);PHAp::mec-12* ($n = 14$) and *mec-12(e1605);PHBp::mec-12* ($n = 11$) males. **b** Tail-touch responses of wild-type ($n = 15$), *tmc-1(ok1859)* ($n = 15$) and *tmc-1(ok1859);PHAp::mec-12* ($n = 13$) hermaphrodites. **c** Tail-touch responses of wild-type ($n = 16$), *tmc-1(ok1859)* ($n = 16$) and *tmc-1(ok1859);PHCp::tmc-1* ($n = 15$) hermaphrodites. The response index represents an average of the forward responses (scored as responded or not responded) in five assays for each animal. We performed a Kruskal-Wallis test followed by a Dunn's multiple comparison test for all comparisons. Orange- hermaphrodites, cyan- males. **d** Schematic of the sensory neurons which function in each sex, with the respective sites of action of *tmc-1* and *mec-12*. All Bar graphs are a box-and-whiskers type of graph, min to max showing all points. The vertical bars represent the median.

AVG, the tail-touch response was rescued when AVG was silenced compared to wild-type males (Fig. 3b). These results suggest that the sexual identity of AVG and consequently its wiring pattern, can shape tail mechanosensory behavior.

Since the behavioral output of harsh touch applied to the tail is a forward movement, we asked whether optogenetic activation of AVG will result in forward movement only in males, as was shown for the sensory phasmid neurons in hermaphrodites[25]. Optogenetic activation of AVG did not affect the forward or total (forward + reverse) speed of the animals in both sexes (Supplementary Fig. 4). However, optogenetic inhibition of AVG reduced the total speed of males only (Fig. 3c, d). Thus, AVG is required for locomotion in a sexually dimorphic manner.

We next tested whether the sex-shared interneurons DVA and PVC also have a dimorphic role in tail mechanosensation. As we were unable to generate a PVC-specific driver, in accordance with previous observations[51] (Supplementary Fig. 5), we focused on the role of DVA. A tail-touch assay on DVA-silenced animals revealed that only hermaphrodites are affected (Fig. 3a, Supplementary Fig. 3c), in agreement with the predicted connectivity (Fig. 1a). Taken together, our results uncover the sex-specific use of different interneurons in the circuit for tail mechanosensation, and assign a functional role for AVG in locomotion and mechanosensation.

Since we uncovered sexually dimorphic functions of the interneuron level in the circuit, we explored potential molecular mechanisms that might govern these differences. To this end, we screened a list of glutamate receptor genes expressed in the relevant interneurons (AVG, DVA and PVC[34,52]; Fig. 3e), as the sensory neurons required for tail mechanosensation are glutamatergic[53]. We found that two glutamate receptors are required for tail mechanosensation in a sexually dimorphic manner: *glr-1* (AMPA type) is needed only in hermaphrodites, while *nmr-1* (NMDA type), which operates in both sexes, has a stronger effect in males (Fig. 3f–h, Fig. 4a). Taken together, our data suggest that the dimorphic nature of the tail mechanosensation circuit spans beyond mere neuronal connectivity to include also different receptor dependency (Fig. 3i).

## NMDA receptor *nmr-1* is required specifically in AVG to mediate tail mechanosensation in males

We next sought to assess the cell-autonomous role of *nmr-1* in tail mechanosensation in AVG. We found that cell-specific expression of *nmr-1* in AVG rescues the defective tail-touch phenotype of mutant males and not of hermaphrodites (Fig. 4a), suggesting that *nmr-1* functions cell autonomously in male AVG. In line with this observation, the expression of *nmr-1* fosmid in AVG was higher in males (Fig. 4b, c).

Since we observed that the sexual identity of AVG is sufficient to determine the behavioral outcome of the circuit when AVG is silenced (Fig. 3b), we asked whether this is also true in *nmr-1* mutant animals. Sex-reversing AVG in *nmr-1* mutant animals switched the tail-touch phenotype to that of the opposite sex, i.e., it reduced the tail-touch response in hermaphrodites with masculinized AVG and enhanced it in

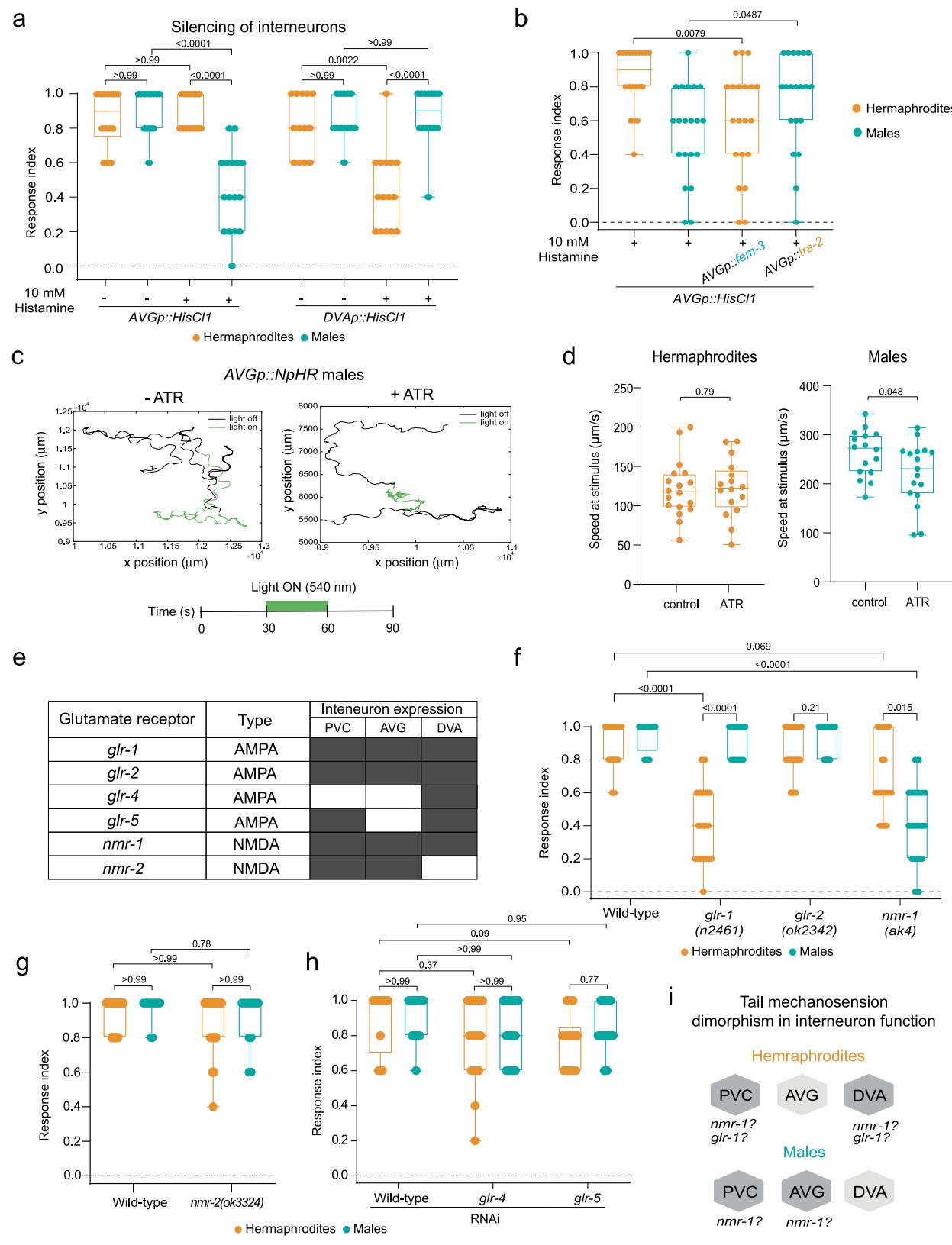

males with feminized AVG (Fig. 4d, e). These results corroborate the observations of the cell-specific rescue experiment, and point to a cell-autonomous role for *nmr-1* in AVG in males that mediates tail mechanosensation. Importantly, the sexual identity of AVG not only dictates the connectivity pattern[43], but also the cell-autonomous molecular pathway that mediates the behavior. Does the sexual identity of AVG modulate the circuit architecture at the sensory level? To test this notion, we checked the tail-touch responses of AVG masculinized hermaphrodites or AVG feminized males in *nmr-1* mutant background, with either PHA or PHC silencing. Remarkably, hermaphrodites with masculinized AVG that show reduced tail-touch responses in the absence of *nmr-1* required PHA and not PHC (Fig. 4f).

**Fig. 3 | Sexually dimorphic integration of tail mechanosensation at the inter-neuron level. a** Tail-touch responses of *AVGp::HisCl1-* and *DVAp::HisCl1*-expressing hermaphrodite and male worms that were tested either on histamine or control plates (see *Methods*). *AVGp::HisCl1* control hermaphrodites: *n* = 14, *AVGp::HisCl1* control males: *n* = 13, *AVGp::HisCl1* histamine hermaphrodites: *n* = 15, *AVGp::HisCl1* histamine males: *n* = 16, *DVAp::HisCl1* control: *n* = 13 animals per group, *DVA-p::HisCl1* histamine hermaphrodites: *n* = 15, *DVAp::HisCl1* histamine males: *n* = 14. **b** Tail-touch responses of *AVGp::HisCl1* hermaphrodites (control and *AVGp::fem-3*) and males (control and *AVGp::tra-2*) that were tested on histamine plates. *n* = 20 animals per group. **c** Representative graph showing the head trajectory (head position) in a control male and ATR-supplemented male. The schematic represents the timeline of the experimental setup. **d** Speed (μm/s) of hermaphrodite and male worms grown on control and ATR plates at the time of light projection. Control hermaphrodites: *n* = 18, ATR hermaphrodites: *n* = 16, control males: *n* = 16, ATR males: *n* = 17. **e** Table listing the glutamate receptors selected for the reverse genetic screen, their expression pattern at the relevant interneurons and their predicted type. Dark boxes represent gene expression. **f**–**h** Tail-touch responses of gene candidates that were examined using mutant strains (**f, g**) or RNAi feeding (**h**) in both sexes. Each experiment was conducted with a control (*him-5(e1490)* (**f**), *him-8(e1489)* (**g**) and *him-5(e1490)* fed with RNAi (**h**)). *him-5(e1490)* hermaphrodites: *n* = 18, *him-5(e1490)* males: *n* = 16, *glr-1(n2461)* hermaphrodites: *n* = 18, *glr-1(n2461)* males: *n* = 16, *glr-2(ok2342)*: *n* = 18 animals per group, *nmr-1(ak4)*: *n* = 20 animals per group, *him-8(e1489)* hermaphrodites: *n* = 20, *him-8(e1489)* males: *n* = 18, *nmr-2(ok3324)* hermaphrodites: *n* = 20, *nmr-2(ok3324)* males: *n* = 18, control for RNAi hermaphrodites: *n* = 13, control for RNAi males: *n* = 12, *glr-4/glr-5* RNAi: *n* = 14 animals per group. **i** Interneuron-level sexual dimorphism of tail mechanosensation with suggested mode of function for *nmr-1* and *glr-1*. The response index represents an average of the forward responses (scored as responded or not responded) in five assays for each animal. In (**d**), we performed a two-sided Mann-Whitney test for each comparison. In (**a, b, f, g, h**) we performed a Kruskal-Wallis test followed by a Dunn's multiple comparison test. All Bar graphs are a box-and-whiskers type of graph, min to max showing all points. The vertical bars represent the median.

On the other hand, males with feminized AVG that show enhanced tail-touch responses in the absence of *nmr-1* do not require PHA or PHC for their responses (Fig. 4f). This result demonstrates that manipulating the circuit organization of the hermaphrodites by masculinizing AVG is sufficient to achieve sensory level processing similar to that of males (requirement of PHA, Fig. 1f). However, in males, switching the sex of AVG is only sufficient to abolish the requirement of PHA at the sensory level. Taken together, the contribution of different sensory neurons in tail mechanosensation is dramatically changed by sex reversal of AVG with a stronger effect in hermaphrodites compared to males.

Our reverse genetic screen uncovered a sex-specific role also for *glr-1* in tail mechanosensation in hermaphrodites (Fig. 3f). However, cell specific rescue experiments of *glr-1* in AVG or DVA did not restore the defective tail-touch response of the hermaphrodites (Supplementary Fig. 6), suggesting *glr-1* operates through a different unidentified interneuron/s.

Our results also show a subtle defect in tail mechanosensation in *nmr-1* mutant hermaphrodites (Fig. 4a, d). Since DVA is required for tail-touch response only in hermaphrodites (Fig. 3a), and *nmr-1* is known to be expressed in hermaphrodites in DVA (Fig. 3e), we checked whether *nmr-1* functions through DVA to mediate tail mechanosensation. Re-expressing *nmr-1* in DVA did not rescue the response of *nmr-1* mutant hermaphrodites (Fig. 4g), suggesting *nmr-1* might be required in a different interneuron, such as PVC, for tail mechanosensation (Fig. 4h).

### Mechanical stimulation of the tail elicits a sexually dimorphic neuronal response in AVG

Since AVG has a role in integrating mechanical information specifically in males, we asked whether it is activated in response to the application of mechanical force to the tail, and whether such activation occurs sex-specifically. We recorded the calcium traces of AVG in both sexes in response to tail mechanical stimulation using a microfluidic device that was adjusted to fit the male body[54] (Supplementary Fig. 7). Similar to previous observations in touch receptor neurons[54], we found that AVG exhibits blue-light-evoked Ca²⁺ transients even in the absence of a mechanical stimulus, suggesting a stimulatory effect of LITE-1 on AVG (Supplementary Fig. 8). We therefore measured the mechanosensitive activity of AVG under a *lite-1* mutant background. Three consecutive tail mechanical stimulations, but not posterior stimulations (mock), elicited neuronal responses in AVG (Fig. 5a–c, Supplementary Movie 1). Importantly, these responses were sexually dimorphic, being significantly lower in hermaphrodites compared to males (Fig. 5a, b; Supplementary Fig. 9a). These findings support our behavioral results, and further indicate that AVG integrates mechanosensory information in a dimorphic manner. In line with the cell-autonomous role we uncovered for *nmr-1* in AVG in tail mechanosensation, we found that AVG responses to mechanical stimulations were reduced in *nmr-1* mutant males compared to controls (Fig. 5d, Supplementary Fig. 9b).

We next asked how the sensory processing of mechanical stimulation is translated at the interneuron level. To explore this issue, we recorded the calcium traces of AVG in response to tail mechanical stimulation in *mec-12* mutant males, where sensory processing of mechanical stimulation is compromised (Fig. 2). Interestingly, *mec-12* mutant males showed significantly lower AVG responses compared to wild-type (Fig. 5f, Supplementary Fig. 9c). This result indicates that proper sensory perception of mechanical stimulation through *mec-12* is critical for the integration at the interneuron level. We also observed lower AVG responses both in *nmr-1* and *mec-12* mutant hermaphrodites, suggesting a role for the two genes in AVG integration of mechanical stimulation in hermaphrodites as well (Fig. 5e, g). We then hypothesized that if *mec-12* re-expression in PHA rescues the behavioral phenotype (Fig. 2a), it would also rescue the neuronal activity of AVG in males. Indeed, re-expression of *mec-12* in PHA was sufficient to restore the neuronal activity in AVG (Supplementary Fig. 9d, e). Taken together, AVG integrates tail mechanosensation in a sexually dimorphic manner, and requires *mec-12*- input from PHA and the glutamate receptor *nmr-1* for this purpose.

### TMC-1 over-expression in PHA reshapes the circuit for tail mechanosensation

To further support a functional role for *tmc-1* in tail touch sensation, we imaged AVG neuronal activity in hermaphrodites in which we over-expressed *tmc-1* in PHA. In hermaphrodites, PHA is connected directly to AVG and both neurons are not normally required for tail mechanosensation (Figs. 1a, b, 3a). We hypothesized that *tmc-1* over expression in PHA would be able to increase the neuronal response of AVG in hermaphrodites. Indeed, over-expressing *tmc-1* in PHA in hermaphrodites resulted in a robust response to tail mechanical stimulation in AVG which was not observed in wild-type hermaphrodites (Fig. 6a). We next wondered whether this ectopic activation of AVG in response to tail touch in PHA::*tmc-1* hermaphrodites is translated to a rescue of the behavioral response of *tmc-1* mutants, despite the fact that normally hermaphrodites do not require PHA for mechanosensation (Fig. 1). Expressing *tmc-1* specifically in PHA resulted in partial but significant rescue of the defective tail-touch responses of *tmc-1* mutant hermaphrodites (Fig. 6b). This result shows that high expression of *tmc-1* in the tail has the power to rearrange the hermaphrodite circuit to activate AVG and elicit a behavioral response, thus supporting an instructive role for *tmc-1* in tail mechanosensation.

### *nmr-1* and *mec-12* play a role in the male mating behavior

The male tail bears the copulatory apparatus and contains specialized sensory structures required for mating[55]. We thus wondered whether the male-specific use of particular neurons and genes for tail mechanosensation may reflect a broader role they have in the mating circuit. To test this, we performed mating assays on mutants for the

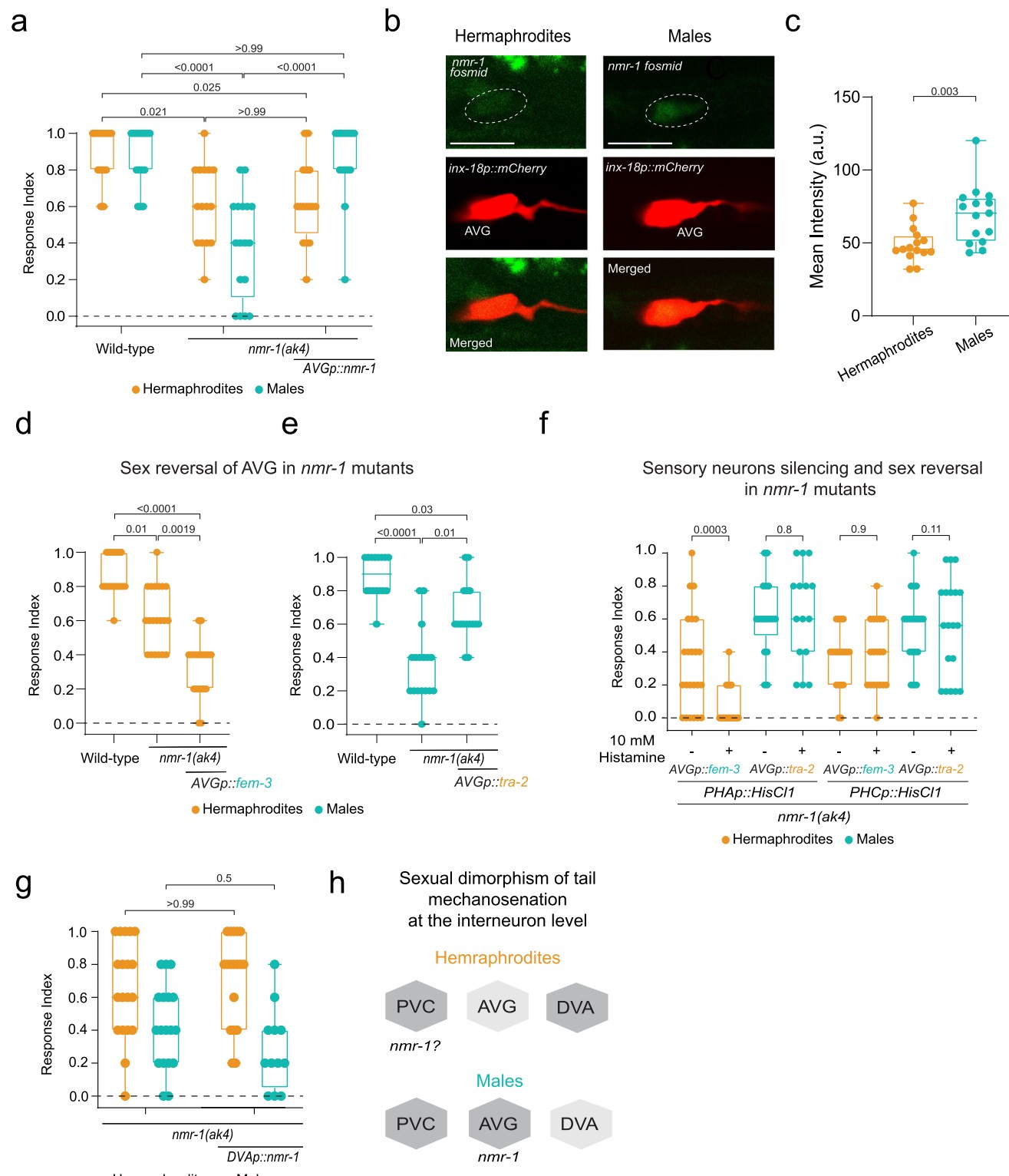

male-specific genes we discovered, *mec-12* and *nmr-1*. While *mec-12* mutant males did not show any defects in mating, *nmr-1* mutant males were defective in their response to contact with hermaphrodites, and in their ability to locate the hermaphrodite vulva, crucial steps in the mating sequence (Fig. 7a, b). Moreover, double mutant males for *nmr-1* and *mec-12* were defective also in the time that passed until successful mating (Fig. 7c), revealing a synthetic interaction between the two genes that enhances the mating defects. These findings demonstrate that *nmr-1* and *mec-12* work in tandem in males to mediate optimal mating, and suggest that the same mechanisms that mediate tail

mechanosensation are also at play in executing appropriate mating behavior.

We therefore tested whether the tail mechanosensation circuit is already functional at earlier stages, even before maturation of the mating circuit of males. We performed tail-touch assay on juvenile wild-type animals and found that the response of both sexes is very low at this stage, compared to adult animals (Supplementary Fig. 10). In addition, juvenile responses of both sexes were unaffected by mutations in the sex-specific genes required later for adult tail mechanosensation i.e. *nmr-1, mec-12* and *tmc-1* (Supplementary Fig. 10). These

**Fig. 4 | Cell-autonomous and sex-specific role for *nmr-1* in tail mechanosensation.**
**a** Tail-touch responses of wild-type ($n = 14$ hermaphrodites, $n = 15$ males), *nmr-1(ak4)* ($n = 15$ hermaphrodites, $n = 17$ males) and *nmr-1(ak4);AVGp::nmr-1* ($n = 16$ hermaphrodites, $n = 15$ males) in both sexes. **b** Representative confocal micrographs of *nmr-1::GFP* fosmid in the AVG interneuron, identified by the expression of mCherry in both sexes. Scale bar is 10 μm. **c** Quantification of (**b**). a.u., arbitrary units. $n = 15$ animas per group. **d, e** Tail-touch responses of hermaphrodites with AVG masculinized hermaphrodites ($n = 20$) (**d**) and AVG feminized males ($n = 16$) (**e**) in *nmr-1(ak4)* mutant background with respective controls: wild-type hermaphrodites: $n = 18$, *nmr-1(ak4)* hermaphrodites: $n = 20$, wild-type males: $n = 16$, *nmr-1(ak4)* males: $n = 17$. **f** Tail-touch responses of AVG masculinized hermaphrodites and AVG feminized males in *nmr-1(ak4)* mutant background expressing either *PHAp::HisCl1* or *PHCp::HisCl1* that were tested either on histamine or control plates (see *Methods*). Animals for each group were collected from two independent

experiments. AVG masculinized hermaphrodites with *PHAp::HisCl1: n = 23* for control group and $n = 22$ for histamine group, AVG feminized males with *PHAp::HisCl1: n = 17* for control group and $n = 15$ for histamine group, AVG masculinized hermaphrodites with *PHCp::HisCl1: n = 19* animals per group, AVG feminized males with *PHCp::HisCl1: n = 19* animals per group. **g** Tail-touch responses of *nmr-1(ak4)* ($n = 19$ hermaphrodites, $n = 18$ males) and *nmr-1(ak4);DVAp::nmr-1* ($n = 18$ hermaphrodites, $n = 12$ males) in both sexes. **h** Receptor site-of-action for tail mechanosensation. The response index represents an average of the forward responses (scored as responded or not responded) in five assays for each animal. In (**c** and **f**), we performed a two-sided Mann-Whitney test for each comparison. In (**a**, **d**, **e**, **g**), we performed a Kruskal-Wallis test followed by a Dunn's multiple comparison test for all comparisons. Orange- hermaphrodites, cyan- males. All Bar graphs are a box-and-whiskers type of graph, min to max showing all points. The vertical bars represent the median.

## Discussion

In this study, we present a unique example for a simple sensory circuit that displays extensive sexual dimorphism, engaging different combinations of cells, connections and molecular pathways in the two sexes to mediate the same sensory modality, i.e., tail mechanosensation (Fig. 8). While very few studies have analyzed the dimorphic properties of sensory circuits in detail in vertebrates, several examples exist in *C. elegans* for either cellular or molecular sensory dimorphism. For example, hermaphrodites detect food-related olfactory cues better than males due to enhanced expression of the odorant receptor ODR-10 in the AWA neuron[1]. Conversely, in the anterior nociceptive circuit, sensory detection of aversive cues seems identical in the two sexes, but the downstream connectivity to interneurons is highly dimorphic[10]. These topographical differences drive sexually dimorphic behavioral responses to nociceptive cues. Notably, in these examples, the dimorphic properties of the circuit lead to dimorphic behavioral outputs, whereas in the circuit we studied here, wild-type males and hermaphrodites respond to the stimulus in the same manner, suggesting that this modality is crucial for the survival of both sexes. It is tempting to speculate that one key driver for the observed dimorphisms in this circuit is the need for males to integrate the mating circuit into the shared nervous system, forcing the redistribution of cells and genes in a different configuration from that of hermaphrodites.

We found four genes that function sex-specifically in the circuit: *mec-12* and *tmc-1* in sensory cells, and *glr-1* and *nmr-1* in interneurons. Previous reports indicate that the alpha-tubulin MEC-12 is required with the beta-tubulin MEC-7 for 15-protofilament microtubule assembly in touch receptor neurons (TRNs), responsible for the transduction of gentle touch[38,56]. Our data suggests that MEC-12 functions through PHA to mediate tail mechanosensation in males, but whether this function requires MEC-7 remains to be determined. The involvement of PHA a ciliated neuron, in tail mechanosensation in males and not the non-ciliated PHC suggests that the defects in *mec-12* mutant males are likely due to aberrant cilia formation in these neurons. It appears that MEC-12, a molecule required for gentle touch[22], is also required for harsh touch, suggesting some overlap between the molecular mechanisms mediating the two modalities, as was previously shown for MEC-10 and MEC-3[23,24].

We also found that the ion channel TMC-1 is a hermaphrodite-specific mediator of tail mechanosensation. This finding adds to a growing repertoire of sensory functions attributed to *tmc-1* in *C. elegans*, including salt sensation, avoidance of noxious alkaline environments, egg laying, gentle-nose touch response, and the inhibition of egg-laying in response to a harsh mechanical stimulus[32,39,57–59]. Taken together, TMC-1 appears to function as a

polymodal ion channel, enabling the processing of different types of information. In *Drosophila*, *tmc* is required for mechanosensitive proprioception[60], whereas in mice, *Tmc1* is expressed in cochlear hair cells and is required for proper mechanotransduction[40,41,61,62]. In both mice and humans, dominant and recessive mutations in *Tmc1* leading to deafness have been identified[61,62]. Sex differences in hearing loss have been documented in mice[63–66], raising the possibility that sex-specific mechanisms involving *Tmc1* may also exist in vertebrates. While a possible dimorphic role for *Tmc1* in higher organisms is yet to be discovered, our results highlight the importance of research on sex-differences for future sex-specific therapeutic approaches.

Our previous and current work together show that AVG is a key dimorphic hub interneuron, responding differently in males and hermaphrodites to at least two separate modalities, tail mechanosensation and nociception[67]. The response of AVG to mechanical stimulation is achieved only when the tail is stimulated but not when other posterior body areas are touched, emphasizing that the input arrives from the tail sensory neurons and that the response to a harsh touch in the tail and the posterior body are distinct modalities involving different cellular and molecular mechanisms[25]. Blue-light-evoked activity in AVG independently of mechanical stimulation suggests that LITE-1 has a role in or upstream to AVG. Supporting this possibility are the findings that AVG is the neuron with the strongest expression of *lite-1*[34] and that PHA requires LITE-1 for $H_2O_2$ sensation[68].

While most studies focus on sex differences between sensory neurons and motor neurons, we emphasize the importance of interneuron plasticity by exposing the intricate use of sex-shared interneurons as dimorphic hubs for different circuits. For example, the AVG interneuron is necessary only in males both for tail-touch and for locomotion speed, whereas the DVA interneuron is necessary for tail-touch only in hermaphrodites and for locomotion speed only in males[69]. Our work also reveals some overlap between the male mechanosensory circuit and the mating circuit, suggesting that males utilize similar mechanisms to execute different behaviors, both in terms of the cells involved (e.g., AVG and PHB in this study, and see also[43]) and the molecular mechanisms (*nmr-1* and partially *mec-12* in this study).

The published connectivity maps suggest that PVC probably plays a critical role in the integration of mechanosensation in both sexes, together with AVG in males and DVA in hermaphrodites. Therefore, a combinatorial system of interneurons seems to be at play to mediate multiple different circuits in each sex.

Despite the superficial similarity in the response of both sexes to touch, we uncover here that the propagation of mechanosensory information is dramatically different, providing evidence for a sexually dimorphic integration of touch signals. Our results contribute to the understanding of how sexually dimorphic circuits function to provide the organism with both the sensation of the environment and the behaviors that ensure its fitness.

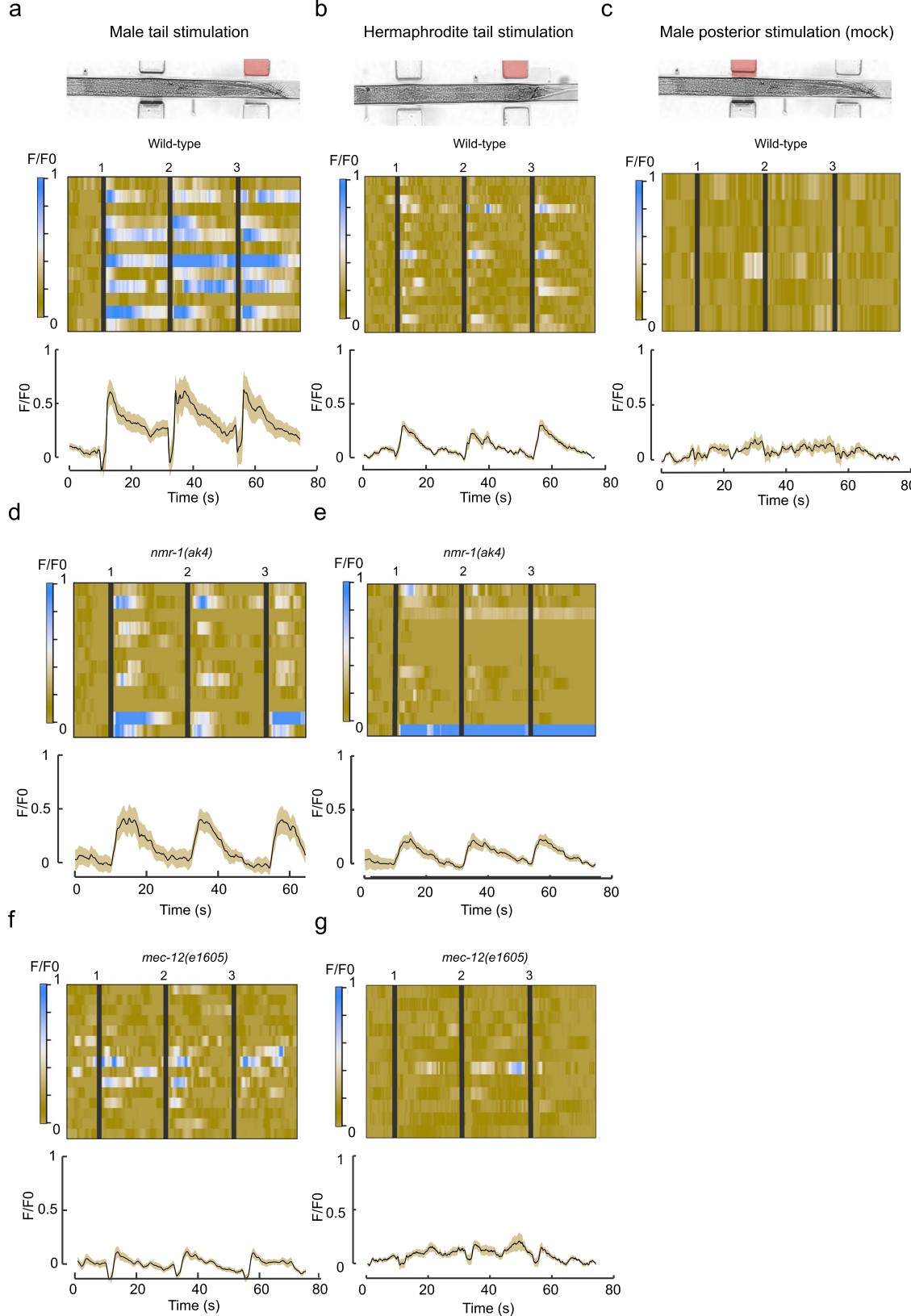

**Fig. 5 | Sexually dimorphic responses of AVG to mechanical stimulation.**
**a**–**c** AVG GCaMP6s calcium responses of males ($n = 13$ recordings from 7 animals) and hermaphrodites ($n = 17$ recordings from 13 animals) to three consecutive tail mechanical stimulations (**a**, **b**) and of males to three consecutive mechanical stimulations anterior to the tail ($n = 6$ recordings from 6 animals) (**c**). Stacked kymographs represent the GCaMP intensity vs. time of individual recordings. Graphs represent average and SD traces of AVG calcium responses. Black vertical lines represent the time when a stimulus was applied. **d**, **e** AVG GCaMP6s calcium responses of *nmr-1(ak4)* mutant males ($n = 13$ recordings from 9 animals) (**d**) and hermaphrodites ($n = 14$ recordings from 8 animals) (**e**) to three consecutive tail mechanical stimulations. **f**, **g** AVG GCaMP6s calcium responses of *mec-12(e1605)* mutant males ($n = 28$ recordings form 16 animals) (**f**) and hermaphrodites ($n = 12$ recordings from 10 animals) (**g**) to three consecutive tail mechanical stimulations. Each 3.5 bar stimulus was applied for two seconds (see *Methods*). Full statistical analysis can be found in Supplementary Fig. 9.

## Methods

### C. elegans strains

Wild-type strains were *C. elegans* variety Bristol, strain N2. *him-5(e1490)* or *him-8(e1489)* were treated as wild-type controls for strains with these alleles in their background. Worms were maintained according to standard methods[70]. Worms were grown at 20 °C on nematode growth media (NGM) plates seeded with bacteria (*E. coli* OP50) as a food source. All the transgenic animals used in this study are listed in Supplementary Table 1, ordered by figures.

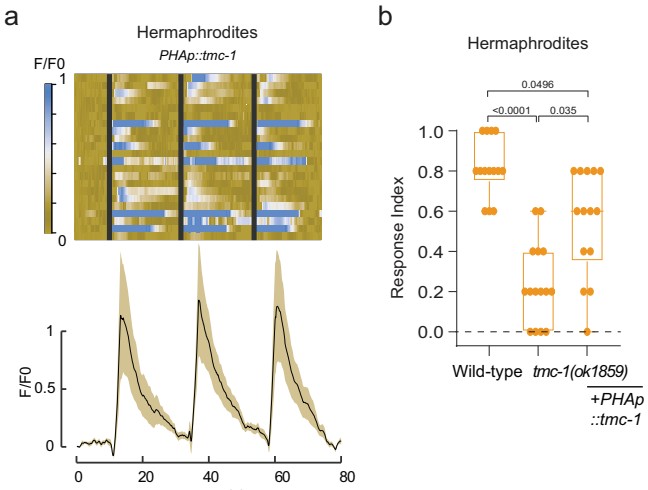

**Fig. 6 | Over-expression of TMC-1 in PHA can elicit a mechanosensory response in hermaphrodites. a** AVG GCaMP6s calcium responses of hermaphrodites with *PHAp::tmc-1* to three consecutive tail mechanical stimulations. Stacked kymographs represent the GCaMP intensity vs. time of individual recordings. Graphs represent average and SD traces of AVG calcium responses. Black vertical lines represent the time when a stimulus was applied. *n* = 23 recordings from 15 animals. Full statistical analysis can be found in Supplementary Fig. 9. **b** Tail-touch responses of wild-type (*n* = 14), *tmc-1(ok1859)* (*n* = 15) and *tmc-1(ok1859)*; *PHAp::tmc-1* (*n* = 14) hermaphrodites. We performed a Kruskal-Wallis test followed by a Dunn's multiple comparison test. Bar graph is a box-and-whiskers type of graph, min to max showing all points. The vertical bars represent the median.

### Histamine-induced silencing

NGM-Histamine plates were prepared as described[30]. 10 mM NGM-histamine was prepared by adding 5 mL of 1 M histamine dihydrochloride to 500 mL of the agar while stirring. NGM-histamine and control-agar (with no addition of histamine) were then poured into labeled petri dishes. NGM-histamine (10 mM) and control plates were stored at 4 °C for no longer than 2 months. Histamine plates were tested using worms that carry a transgene with a pan-neuronal HisCl1 (*tag-168::HisCl1::SL2::GFP*)[30]. After a few minutes on histamine plates, these worms were paralyzed completely, validating the potency of the histamine plates.

### Tail-touch assay

The assay was based on[25]. In brief, L4 animals were isolated the day before the experiment and stored at 20 °C overnight. On the day of the experiment, single 1-day adult animals were transferred into NGM plates freshly seeded with 30 μl OP50. After 30 min of habituation, animals were tested by applying a touch to the tail with a flattened platinum wire pick. Touch was applied to worms that did not move or moved very slowly. Each worm was tested five times with intervals of at least 10 s between each trial. The tested worm was given a score of one if it moved forward in response to the touch, and zero if it did not move forward. The response index was then calculated as the average of the forward responses. For assays using histamine-gated chloride channels, animals were grown on histamine plates or control plates seeded with 200–300 μl OP50 overnight. On the day of the experiment, single 1-day adult animals were assayed on histamine plates or control plates seeded with 30 μl OP50. For assays using RNAi, *him-5(e1490)* animals were grown on RNAi plates or control plates seeded with 200 μl of the RNAi bacteria. For assays using juvenile animals, worms were bleached and grown at 25 °C for one day to reach L3. The assay was done at L3 stage and animals were kept on the assay plate and scored for the sex when they reached L4. Experiment was done until a sufficient number of males (13 or more) was collected, and statistics was done with an equal number of hermaphrodites from the same experiment that were chosen randomly. All statistical analysis for behavioral assays were computed using GraphPad version 9.

### Molecular cloning

To generate the *mec-12* rescue constructs pYS55 and pYS57 (*srg-13p::mec-12* for PHA rescue and *gpa-6::mec-12* for PHB rescue, respectively), *mec-12* cDNA was amplified from a N2 mixed-stage cDNA library

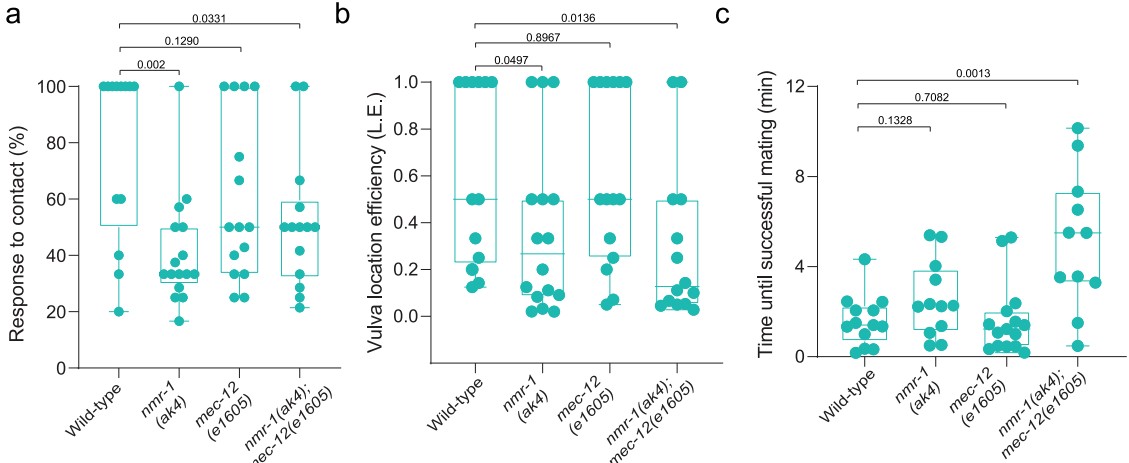

**Fig. 7 | *nmr-1* and *mec-12* function in the male mating behavior. a** Quantification of response to contact in wild-type (*n* = 13) *nmr-1(ak4)* (*n* = 16), *mec-12(e1605)* (*n* = 15) and *nmr-1(ak4)*;*mec-12(e1605)* (*n* = 14) males. **b** Quantification of vulva location efficiency (L.E., see Methods) in wild-type (*n* = 13) *nmr-1(ak4)* (*n* = 16), *mec-12(e1605)* (*n* = 15) and *nmr-1(ak4)*;*mec-12(e1605)* (*n* = 14) males. **c** Quantification of the time until successful mating. in wild-type (*n* = 13) *nmr-1(ak4)* (*n* = 12), *mec-12(e1605)* (*n* = 15) and *nmr-1(ak4)*;*mec-12(e1605)* (*n* = 11) males. We performed a two-sided Mann-Whitney test for each comparison. All Bar graphs are a box-and-whiskers type of graph, min to max showing all points. The vertical bars represent the median.

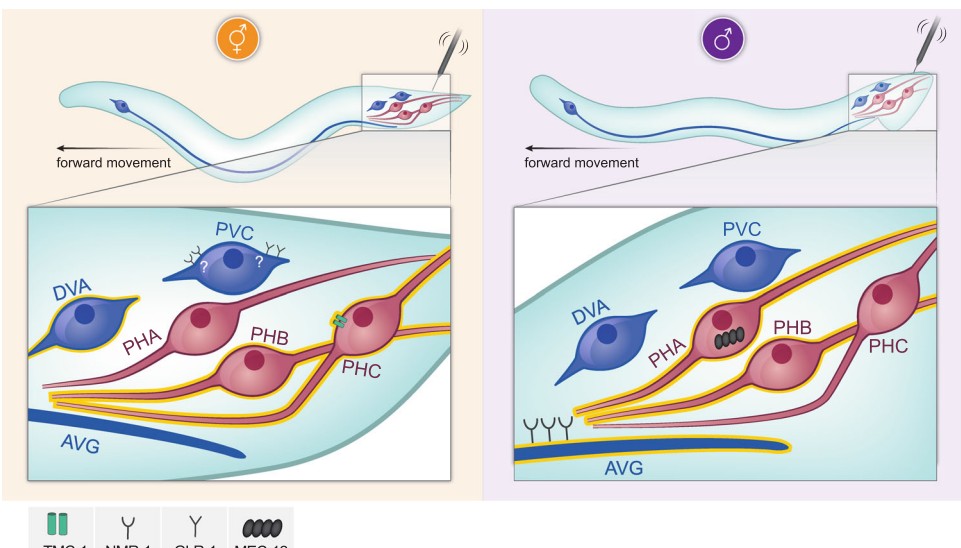

**Fig. 8 | Sexually dimorphic cellular and molecular mechanisms controlling tail mechanosensation.** A model depicting the cellular and molecular elements mediating tail mechanosensation in each sex. Neurons that function in each sex are highlighted in yellow. Labeling of TMC-1, MEC-12, NMR-1 and GLR-1 describes their suggested functional role in these cells. Figure was made by Weizmann Institute's graphics unit.

and cloned by Gibson assembly[71] into *srg-13p* or *gpa-6p* plasmid backbones.

To generate the *mec-12* rescue construct pHS16 (*che-12p::mec-12*), *mec-12* cDNA was amplified from a N2 mixed-stage cDNA library and cloned by Gibson assembly[71] into a *che-12p* plasmid backbone.

To express *tmc-1* cDNA specifically in PHC, *tmc-1* cDNA was amplified from pEK158 (kind gift from Eva Kaulich and Bill Schafer) and fused to a plasmid backbone containing the *eat-4p11* promoter and upstream of an *SL2::NLS::tagRFP* cassette by Gibson assembly, creating pYS54. The *tmc-1 cDNA::SL2::NLS::tagRFP* was then subcloned under the *srg-13* promoter by Gibson assembly to generate the PHA rescue plasmid pYS56.

To generate the optogenetics plasmids (pHS5 - *inx-18p::NpHR::mCherry* and pHS6 - *inx-18p::Chr2::mCherry)*, NpHR and Chr2 were amplified from plasmids provided by Alon Zaslaver and cloned into pMO46[43].

To generate the *nmr-1* rescue constructs pHS14 (*inx-18p::nmr-1* cDNA) and pYS53 (*WT300::pes-10::nmr-1* cDNA), *nmr-1* cDNA was amplified from a N2 mixed-stage cDNA library and cloned by Gibson assembly into an *inx-18p* plasmid backbone, and pHS11 (*WT300::pes-10::mCherry*, swapping the mCherry fragment), respectively. WT300 was amplified from genomic DNA[72]. To generate pYS52 (*WT300::pes-10::HisCl1::SL2::GFP*), *HisCl1::SL2::GFP* was amplified from pMO46[43] and cloned into pHS11 (*WT300::pes-10::mCherry*, swapping the mCherry fragment) using Gibson assembly.

pYS56 *(WT300::pes-10::glr-1* cDNA) was created by amplifying *glr-1* cDNA from a N2 mixed-stage cDNA library and cloned by Gibson assembly into pYS53 (*WT300::pes-10::nmr-1*), swapping nmr-1 with glr-1 cDNA.

The translational *mec-12p::mec-12::GFP* construct was created by PCR fusion[73]. A 10.6Kb *mec-12* genomic fragment encompassing 3Kb upstream of TSS and the full *mec-12* cDNA was amplified from N2 genomic DNA and connected by PCR fusion to a pPD95.75-based GFP::unc-54 3'UTR fragment.

## RNAi
RNA interference was performed using the feeding method[74]. L4 hermaphrodites were fed HT115 bacteria carrying dsRNA for the relevant genes or a control empty vector, and their progeny were assayed for a tail-touch response at 1-day adult. *pos-1* RNAi served as a positive control for RNAi efficiency. A fresh *pos-1* control was prepared for each RNAi experiment.

## DiD staining
Worms were washed with M9 buffer and incubated overnight in 1 ml M9 and 5 µl DiD dye (Vybrant™ DiD Cell-Labeling Solution, Thermo-Fisher) at -10–20 rpm. The worms were then centrifuged and transferred to a fresh plate.

## Microscopy
Animals were mounted on a 5% agarose pad on a glass slide, on a drop of M9 containing 100–200 mM sodium azide (NaN3), which served as an anesthetic. A Zeiss LSM 880 confocal microscope was used with 63x or 10x magnification. Zen software was used for imaging (version 2.3). For *nmr-1* fosmid, AVG was identified using *mCherry* expression, and the z-plane with the strongest signal was chosen to measure the fluorescence intensity using ImageJ version 1.52p. All statistical analysis imaging were computed using GraphPad version 9.

## Optogenetics
For optogenetic inhibition or activation, we used worms expressing NpHR or Chr2, respectively, only in AVG (*inx-18p::NpHR::mCherry/inx-18p::Chr2::mCherry*). L4 worms were picked a day before the experiment and separated into hermaphrodite and male control and experiment groups. They were transferred to newly seeded plates with 300 µl OP50 that was concentrated 1:10. ATR (all-trans-retinal) was added only to the experimental groups' plates, to a final concentration of 100 µM. As ATR is sensitive to light, all plates were handled in the dark. Tracking and optogenetics was done on unseeded NGM plates. On the day of the experiment, the plates were seeded with 30 µl OP50, and ATR was added to the experimental group's plates. To keep the worms in the camera field of view, a plastic ring was placed on the agar and a single worm was placed inside the ring. After 10 min of habituation, the worm was tracked for 90 s, with 30 s of 540 nm LED activation initiated 30 s after tracking commencement. The LED intensity was 0.167 mW/mm2. Speed measurements were extracted for the different time periods from WormLab (MBF Bio-science[75]) using "label-analysis". The head position trajectory quadrants were also extracted from WormLab, and inserted into a graph using MATLAB (version R2019B). Statistical were computed using GraphPad version 9.

For optogenetic activation, 3–5 worms were placed inside a plastic ring on the agar each time. After 10 min of habituation, the worms were tracked for 60 s, with a 2-second blue-light stimulus introduced every 10 s (5 stimuli in total). The LED intensity was 1.47 mW/mm2. Speed measurements were extracted for all the time points from WormLab. For each stimulus, the mean total speed was calculated before the stimulus (3 s) and after the stimulus (5 s). Calculations were done using MATLAB (version R2019B).

## Calcium imaging of animals in the microfluidic chip

**Mold design and preparation.** Microfluidic devices are based on the body wall touch device[54]. Subtle modification in the trapping channel (width and height have been adjusted to 30 μm) were applied to fit male animals, which are skinnier than hermaphrodites, and the devices were cast with a PDMS base polymer/curing agent ratio to allow for large deflection of the diaphragm. The male body wall touch design, drawn with AutoCAD 2021, is illustrated in Supplementary Fig. 7a; its file is available to download in the Supplementary Material (Supplementary Data 1). Supplementary Fig. 7a shows how perfectly the modified trapping channel fits the males. To further enable a larger deflection, the size of the diaphragm was decreased to 10 microns. The deflection of the actuator under different pressure levels is presented in Supplementary Fig. 7c; the graph highlights the consistency of the numerical results with the experimental data.

The mold was prepared using a standard soft-lithography technique in a class ISO2 cleanroom[76]. In short, 4-inch silicon wafers were cleaned in Piranha and dehydrated by baking at 150 °C. The dried, cleaned wafers were spin-coated with 1 ml/inch SU8-50 (MicroChem, Newton Massachusetts, USA) to obtain a 30-micron height (500 RPM (100 RPM/s acceleration for 15 s)) and then ramped to a final speed 3300 RPM for 45 s (300 RPM/s acceleration). The SU8 layer was pre-baked at 65 °C for 5 min and then at 95 °C for 15 min. The device design was 'printed' with a UV laser onto the SU8 substrate with a mask-less aligner (MLA, Heidelberg Germany). Once the printing process was finished, the wafer was post-baked for 1 min at 65 °C and then for 4 min at 95 °C. The exposed SU8 was developed for 6 min in an SU8 developer (MicroChemicals) and rinsed with propanol, to remove unexposed resin. Thereafter, the wafer was hard-baked for 2 h at 135 °C and used for replica-molding PDMS devices. First, the surface of the structured wafer was vapor-phase silanized with Chlorodimethylsilane (Sigma-Aldrich, Missouri, United States) to prevent adhesion of the PDMS to the substrate and facilitate lift-off of the PDMS during the peeling process. The PDMS base polymer and curing agent were mixed at a ratio of 15:1 and approximately 40 mL per wafer were prepared and degassed before pouring to prevent bubbles. The degassed PDMS was poured into the mold and cured in the oven at 85 °C for two hours. After lift-off and trimming the devices to fit onto standard #1.5 cover glasses, all inlets and outlets were punched with a biopsy punch (0.75 mm), which served as a receptacle for the connection tubes. Finally, to bind the PDMS with the cleaned cover slip, the surface of the coverslip was activated by a plasma treatment (Plasma Asher PVA TePla 300), and to increase the bonding quality, the bonded chip was placed on a hotplate (120 °C) for 10 min. The punch holes were plugged with SC22/15-gauge metal tubes connected to 22-gauge catheter tubing.

**Animal loading, stimulation and fluorescence microscopy.** Animals were placed in the device exactly as described and shown in[77]. In brief, 3–4 synchronized day-one adult animals (males or hermaphrodites) were picked from an NGM plate containing OP50 bacteria and transferred into a 15 μL droplet of M9 buffer. Using a stereo dissecting scope (Leica S9), the animals were aspirated into a SC22/15-gauge metal tube (Phymep) connected to a 3 mL syringe (Henke Sass Wolf) with a PE tube (McMaster-Carr) pre-filled with M9 buffer. The loading tube was inserted into the inlet port of the device, while gentle pressure to the

plunger of the syringe released the animals into the loading chamber. The microfluidic chip was then transferred to a compound fluorescence microscope (Leica DMi8), and with the 25 × 0.95 water immersion lens, the animal was positioned such that the tail aligned with an actuator (Fig. 5a), ready to accept a mechanical stimulus. Once positioned properly, AVG was brought into focus and the stimulation protocol was started. GCaMP in AVG was excited with a cyan LED of a Lumencor SpectraX light engine (470 nm @ 2% transmission or 0.32 mW at the sample plane). The incident power of the excitation light was measured with a microscope slide powermeter head (Thorlabs, S170C) attached to PM101A power meter console (Thorlabs). Emission was collected through a 515/15 nm emission filter (Semrock) and videos were captured at a 10 Hz acquisition rate (85 ms exposure time) with a Hamamatsu Orca Flash 4.3 for 65 or 75 s, using HCImage software. A master-pulse was used to synchronize the camera acquisition with the light exposure. Likewise, the camera SMA trigger was connected to the OB1 pressure controller and used to synchronize the image acquisition with the pressure protocol through the ElveFlow sequencer software prior to the imaging routine, ensuring precise and repetitive timing of the pressure protocol. The sequence consisted of 100 pre-stimulus frames and three consecutive mechanical stimuli, each separated by 20 s. The stimulus profile consisted of a 2-second 250 kPa pressure step, overlayed with a 150 kPa oscillation, resulting in a maximum deflection of 9 μm (Supplementary Fig. 7c).

**Analysis.** Image processing and analysis were performed as described in[20]. In brief, to extract GCaMP intensity over time, images were pre-processed in ImageJ and then imported into python to extract signal intensity using in-house procedures written in Python (Version 2.7) or IgroPro (Version 6.370). In short, the image sequence was cropped to a small area surrounding the cell body (or the axon, Supplementary Fig. 9d) of the neuron of interest and a Gaussian filter was applied. All pixel values in the ROI were ranked and the background intensity was extracted from the 0–10th percentile and the GCaMP signal was extracted from the 90–100th percentile. After background subtraction, signal intensity (F) was normalized to the median value of the first 100 frames (Fb, before the mechanical stimulus was applied according to (F-Fb)/Fb).

To calculate a time-varying p-value comparing two datasets derived from different conditions, we first calculated the average calcium response after performing a non-linear baseline subtraction (bleach correction, drift) of each individual recording, taking advantage of an iterative algorithm to suppress the baseline by means in local windows[78]. To minimize spurious fluctuations in p-values due to imaging noise, a smoothing kernel based on the 2nd derivative penalty, known as a Whittaker smoother[79], was applied. The t-test statistics was then used to calculate the p-value considering the degrees of freedom, $df = N1 + N2-2$; the null hypothesis was rejected if the absolute value of $t$ was larger than its critical value at the level of significance alpha = 0.01. Outliers were identified based on a z-score > |7|, which means that Calcium recordings that appeared different than 7 standard deviation from the mean were omitted from the statistical analysis according to $z = (t-\mu)/\sigma$ (with $z$ as the z score, $t$ as the testable variable, μ and sigma as the mean or the standard deviation of the population). In total 2 outliers in two different data sets have been identified. All operations were performed in R version 4.0.2 (2020-06-22), using custom routines involving the baseline package V1.3-1.

## Mating assay

Mating assays were as described in ref. 43. In brief, Early L4 males were transferred to fresh plates and kept apart from hermaphrodites until they reached sexual maturation. Single virgin males were assayed for their mating behavior in the presence of 10–15 adult *unc-31(e928)* hermaphrodites on a plate covered with a thin fresh *E. coli* OP50 lawn. Mating behavior was scored within a 12-minute time window or until

the male ejaculated, whichever occurred first. Mating was recorded using a Zeiss Axiocam ERc 5 s mounted on a Zeiss stemi 508. The movie sequence was analyzed and the males were tested for their ability to perform an intact mating sequence[80]. Males were scored for their time until successful mating, contact response and vulva location efficiency. Males that did not mate within 12 min were not analyzed for time until successful mating. Contact response requires tail apposition and initiation of backward locomotion. Percentage response to contact = $100 \times$ (the number of times a male exhibited contact response/the number of times the male makes contact with a hermaphrodite via the rays)[43,81]. Vulva location efficiency (L.E.)[82], or the ability to locate the vulva, was calculated as 1 divided by the number of passes or hesitations at the vulva until the male first stops at the vulva or until the end of the recording. All statistical analysis for mating assays were computed using GraphPad version 9.

### Reporting summary

Further information on research design is available in the Nature Portfolio Reporting Summary linked to this article.

## Data availability

The authors declare that all data generated or analyzed during this study are included in this published article (and its supplementary information files). The raw data generated in this study are provided in the Source Data file. Source data are provided with this paper.

## Code availability

The code generated in this study has been deposited at https://doi.org/10.34933/wis.000652 and is publicly available as of the date of publication.

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

## Acknowledgements

We thank members of the Oren-Suissa lab for their critical insights regarding the manuscript. The plasmids to amplify Chr2 and NpHR were a generous gift from Alon Zaslaver. The primer sequences to generate pHS11 were a kind recommendation of Carmine Puckett Robinson. The pEK158 plasmid and the AQ4330 strain were a kind gift from Eva Kaulich and Bill Schafer. Some strains were provided by the CGC, which is funded by the NIH Office of Research Infrastructure Programs (P40 OD010440). This work was supported by the European Research Council ERC-2019-STG 850784 (MOS), Israel Science Foundation grant 961/21 (MOS). MOS is grateful to the Azrieli Foundation for the award of an Azrieli Fellowship, and is the incumbent of the Jenna and Julia Birnbach Family Career Development Chair. MK acknowledges financial support from the ERC (MechanoSystems, 715243), HFSP (CDA00023/2018), Spanish Ministry of Economy and Competitiveness through the Plan Nacional (PGC2018-097882-A-I00), FEDER (EQC2018-005048-P), "Severo Ochoa" program for Centres of Excellence in R&D (CEX2019-000910-S; RYC-2016-21062), Fundació Privada Cellex, Fundació Mir-Puig, and from Generalitat de Catalunya through the CERCA and Research program (2017 SGR 1012).

## Author contributions

H.S., Y.S. conducted and analyzed the experiments, S.K. conducted and analyzed the calcium imaging experiments, E.B.B. contributed to behavioral experiments (tail-touch assays). M.K. and M.O.S. supervised and designed the experiments. H.S., Y.S. and M.O.S. wrote the paper.

## Competing interests

The authors declare no competing interests.
