## [Peer Review File · Nature Communications]

Sexually dimorphic architecture and function of a mechanosensory circuit in *C. elegans*REVIEWER COMMENTS

Reviewer #1 (Remarks to the Author):

This study adds to the growing literature regarding how sexually dimorphic behaviors are generated at the genetic and neuronal circuit levels. In contrast to previous studies that have focused primarily on how sensory neurons respond dimorphically to stimuli, this study shows how function and activity of a sex-shared interneuron gives rise to sexually dimorphic behavioral output. In general, the ways that differences in interneurons can affect circuit functions are often less emphasized than those in sensory or motor neurons, so this study is an exciting investigation of how important interneuron plasticity is to the behavioral output of a circuit.

The only major weakness of this study is the attempts to link expression levels/patterns of relevant genes to dimorphic effects; expression patterns of the genes involved behaviorally are not clearly able to demonstrate why the behavioral effect should be dimorphic (specific cases detailed below), although the fact that the complete connectomes of *C. elegans* are available in both sexes and that the connectivity patterns have been shown to be dimorphic allows an alternate explanation for this. The behavioral and activity-based assays that form the foundation of the study are well-done.

Figure 2: Can fluorescent images of the *mec-12* expression pattern in PHA/PHB be included in addition to the quantification (here or in a supplement)?

Figure 3: The text states that *nmr-1* is required in both sexes but has a stronger effect in males. Panel 3F however shows no difference between wild-type and *nmr-1* mutant hermaphrodites?

Figure 4: The images of the *nmr-1* fosmid don't inspire much confidence that *nmr-1* is indeed expressed in AVG; the region indicated as AVG doesn't look any different from the surrounding area, which has diffuse GFP everywhere. In the image shown the diffuse signal seems overall brighter in males than in hermaphrodites, which makes a fluorescence quantification without any apparent normalization a bit concerning. Perhaps this result could be strengthened by using additional/different reporters, and/or normalizing the expression to something (the *inx-18p::mCherry* shown should work, although in the images selected for the figure the signal is very oversaturated; pixel distortions are visible).

Figure S5: It is questionable to use a heterologous expression construct for GLR-1 to claim a sexual dimorphism. In the absence of any of *glr-1*'s endogenous regulatory information, what does an expression difference even mean? In general, the *glr-1* results are confusing; an effect in hermaphrodites (but not males) that can't be rescued, isn't affected by sex reversal, and by a heterologous expression construct is higher in males than hermaphrodites? The authors propose that

this has something to do with PVC, which can't be assessed individually due to lack of specific promoters. Perhaps a bit drastic, but I would consider removing the *glr-1* results altogether. The paper would be just as good without it, and maybe one day there will be better tools for PVC that would allow the role of *glr-1* to be more carefully assessed.

Minor comments:

I'm curious how some of these results (interneuron silencing and mutants, especially) look in juvenile animals before the mating circuit develops. This probably wouldn't affect the conclusions of the manuscript either way for me so it isn't crucial, but seems like an easy enough experiment and could be fun to see.

I'd revise the concluding sentence in the mating behavior section (lines 335-337); seeing an effect in an *nmr-1* mutant implies that the mating circuit uses glutamatergic neurotransmission, which is interesting but not terribly scandalous, since many circuits and behaviors use glutamatergic neurotransmission. I'm not sure it could be called a "hijacking" to have a circuit that requires glutamate.

In the discussion, the proposal that the sexually dimorphic phenomenon demonstrated in this paper is the result of "extensive sexually dimorphic reorganization during evolution," and "this modality was important enough for both sexes to withstand the evolutionary pressures that changed its constituents" seems to suggest that there was some ancestral state of a non-dimorphic touch circuit that then changed to the observed situation in *C. elegans*. One of the reasons proposed for this change is "the need to integrate the mating circuit," which again suggests that there existed an ancestral non-dimorphic state in an animal/nematode that apparently reproduced without mating, and then mating was integrated as an evolutionary novelty. In the absence of any evolutionary evidence in this paper that such an ancestral state exists (ie comparison to male/female species, which are proposed to be the ancestral state, or to the other instances that hermaphroditism has arisen), this seems very speculative. If literature exists suggesting that there is indeed an ancestral non-dimorphic touch circuit in a non-mating species it could be cited to strengthen this argument, otherwise these claims are a bit distracting.

Reviewer #2 (Remarks to the Author):

This paper dissects the molecular and cellular architecture of a sexually dimorphic circuit for posterior touch in *C. elegans*. A very interesting aspect of the findings is that males and hermaphrodites use different neural strategies to achieve identical behavioural outputs and that while all circuit components are present in both sexes, each sex relies on different subsets of neurons for touch-dependent behavioural responses. Specifically, the authors use cell-specific silencing of neurons to show that males

rely on sensory neurons PHA and PHB and interneuron AVG to elicit forward scape in response to posterior touch whereas hermaphrodites rely on PHC and PHB sensory neurons and DVA interneuron. Analysis and rescue of mutants show that touch sensation requires the function of *mec-12*, likely in PHA and PHB, and *nmr-1* in AVG in males. In contrast hermaphrodites require the function of *tmc-1*, acting in PHA, and *glr-1*. The role of the identified genes and neurons in male touch sensation is further supported with imaging of neuronal activity in AVG. Finally, the authors propose that a potential reason for the circuit dimorphism may be the necessity of males to integrate other (male-specific) circuits and functions into the tail mechanosensory circuit. This is based on the defective mating behaviour of *nmr-1* mutant males and previous studies which show that AVG interneurons are involved in male mating.

Overall, the experiments are convincing, carefully done and support the main conclusions of the paper. In addition, and as mentioned above, the findings are of great interest not only to those working on sexual dimorphism but also to systems neuroscientists, evolutionary biologists and ethologists.

My main concern is the implication of *mec-12* acting in PHA and PHB in males for behaviour and AVG neural activity. The authors use too broad a promoter (*che-12*) for their *mec-12* rescues. However, they do have PHA and PHB specific promoters, which they use for cell silencing. Could they not use those to confirm that *mec-12* is indeed acting in PHA and PHB? It would be good to see this rescue not just for behaviour but also for neuronal activity in AVG.

The site of action of *glr-1* is also not reported. Can they rescue the touch response in hermaphrodites by expressing in DVA? rescue

In addition, to understand the sexually dimorphic circuit architecture it would be informative to see what is the contribution of PHA and PHC to the behaviour observed in AVG sex-reversal experiments. Is the *nmr-1* mutant male enhanced touch response still dependent on PHA? Or is it now dependent on PHC? Can the response of *nmr-1* mutant hermaphrodites be further reduced by silencing PHA? Or does response still rely on PHC?

Minor comment:

In the Discussion, the suggestion of a threshold mechanisms in AVG for behavioural output is not convincing because the results only show that AVG is not required in hermaphrodites for touch response (other neurons may compensate), but AVG may still contribute to behaviour.

Reviewer #3 (Remarks to the Author):

Hagar et al report a sexual dimorphism in a mechanosensory circuit of *C.elegans*. As for tail mechanosensation, it is very interesting. But this study still need more clarification before it can be at the level of Nature Communications. More time is required for specific parts to be commented and debated.

Specific Comments:

1. MEC-12 was assayed in the tail mechanosensation of males but no mating defects which may suggest that MEC-12 is not the primary mechanoreceptor as the study is showing with sexual dimorphism.

2.The touch assay and response index are not enough to support *tmc-1* tail mechanosensation of hermaphrodite. TMC-1 may regulate resting potential, or form a mechanosensitive complex with other proteins. To clarify this, specific rescue, calcium imaging and ephys recording would be good.

3.Why *tmc-1* required in PHC neurons for hermaphrodite tail mechanosensation but not in PHA neurons? PHA are ciliated neurons but not PHC neurons are different in males (striated rootlets) than hermaphrodite.

4.Line 385; "TMC-1 appears to function as a polymodal ion channel, enabling the processing of different types of information". This is an overstatement in the conclusion as the study failed to show more about *tmc-1*. In addition, TMC-1 may not be a sodium-sensitive channel (L381), according to some published papers.

5.Line 178-180; They introduced a sex-determining factor specifically in AVG to switch its sexual identify and connectivity to opposite sex; There no more mention about this "switch" except references.

6.Harsh touch was not assayed and only optogenetics of AVG were showed.

7.AVG evoked Calcium transients in the absence of mechanical stimulus.

ANSWERS TO REVIEWER COMMENTS MANUSCRIPT NCOMMS-22-05322

Reviewer #1 (Remarks to the Author):

This study adds to the growing literature regarding how sexually dimorphic behaviors are generated at the genetic and neuronal circuit levels. In contrast to previous studies that have focused primarily on how sensory neurons respond dimorphically to stimuli, this study shows how function and activity of a sex-shared interneuron gives rise to sexually dimorphic behavioral output. In general, the ways that differences in interneurons can affect circuit functions are often less emphasized than those in sensory or motor neurons, so this study is an exciting investigation of how important interneuron plasticity is to the behavioral output of a circuit.

The only major weakness of this study is the attempts to link expression levels/patterns of relevant genes to dimorphic effects; expression patterns of the genes involved behaviorally are not clearly able to demonstrate why the behavioral effect should be dimorphic (specific cases detailed below), although the fact that the complete connectomes of *C. elegans* are available in both sexes and that the connectivity patterns have been shown to be dimorphic allows an alternate explanation for this. The behavioral and activity-based assays that form the foundation of the study are well-done.

We thank the reviewer for his/her feedback. We edited the text so to avoid conclusions that connect between expression patterns and observed behavioral effects. We also addressed the specific issues listed below.

Figure 2: Can fluorescent images of the *mec-12* expression pattern in PHA/PHB be included in addition to the quantification (here or in a supplement)?

We now include images for the transcriptional *mec-12* reporter in Supplementary Fig. 1. To observe the expression of MEC-12 protein, we generated a translational reporter of *mec-12* (*mec-12p::mec-12::gfp*). We were mostly able to observe the expression of MEC-12 in PHA, and this expression was very low in both sexes. This finding correlates with the *CenGen* dataset, which also shows low transcripts levels of *mec-12* in PHA and PHB compared to other cells (Taylor et al, 2021). This result can be found in Supplementary Fig. 1.

Figure 3: The text states that *nmr-1* is required in both sexes but has a stronger effect in males. Panel 3F however shows no difference between wild-type and *nmr-1* mutant hermaphrodites?

In our hands, a possible mechanosensory defect for *nmr-1* mutant hermaphrodites reached significance in 3 out of 4 separate experiments. Compare the p values of Fig. 3f (0.069) with that of 4a (one comparison between wild-type and *nmr-1* mutant hermaphrodites, $p=0.021$ and between wild-type and mutant *nmr-1* hermaphrodites with rescue of *nmr-1* in AVG, which we have shown that only affects males, $p=0.025$) and 4d ($p=0.01$). To avoid confusions, we added to the text a reference to Fig. 4a (lines 207-210).

Figure 4: The images of the *nmr-1* fosmid don't inspire much confidence that *nmr-1* is indeed expressed in AVG; the region indicated as AVG doesn't look any different from the surrounding area, which has diffuse GFP everywhere. In the image shown the diffuse signal seems overall brighter in males than in hermaphrodites, which makes a fluorescence quantification without any apparent normalization a bit concerning. Perhaps this result could be strengthened by using additional/different reporters, and/or normalizing the expression to something (the *inx-18p::mCherry* shown should work, although in the images selected for the figure the signal is very oversaturated; pixel distortions are visible).

We repeated the experiment using a different line of *nmr-1* fosmid that was less noisy (Fig. 4b-c). The signal in AVG in the new images is much clearer and the result remained the same – higher expression of *nmr-1* in AVG in males compared to hermaphrodites. *nmr-1* expression in AVG has been previously established by Pereira et al eLife 2015 and Taylor et al Cell 2021 (*CenGEN*).

Figure S5: It is questionable to use a heterologous expression construct for GLR-1 to claim a sexual dimorphism. In the absence of any of *glr-1*'s endogenous regulatory information, what does an expression difference even mean? In general, the *glr-1* results are confusing; an effect in hermaphrodites (but not males) that can't be rescued, isn't affected by sex reversal, and by a heterologous expression construct is higher in males than hermaphrodites? The authors propose that this has something to do with PVC, which can't be assessed individually due to lack of specific promoters. Perhaps a bit drastic, but I would consider removing the *glr-1* results altogether. The paper would be just as good without it, and maybe one day there will be better tools for PVC that would allow the role of *glr-1* to be more carefully assessed.

We agree with the reviewer that the heterologous construct of GLR-1 is lacking. Therefore, these results were removed, as well as the sex-reversal results. We carried out an additional experiment (see answers to reviewer #2) to rescue GLR-1 expression in DVA, but we could not rescue the defect in the hermaphrodites. These negative results are only briefly mentioned (lines 262-264) and appear in Supplementary Fig. 6. We think it is important to let the community know which cells were examined in the behavioral assays.

Minor comments:

I'm curious how some of these results (interneuron silencing and mutants, especially) look in juvenile animals before the mating circuit develops. This probably wouldn't affect the conclusions of the manuscript either way for me so it isn't crucial, but seems like an easy enough experiment and could be fun to see.

We performed the tail-touch assay on L3 animals before sexual maturation in the following backgrounds: *wild-type*, *nmr-1(ak4)*, *mec-12(e1605)* and *tmc-1(ok1859)*. Overall, the responses were very low already in wild-type animals (Supplementary Figure 10). Mutant animals in all backgrounds showed similar results, with no sexual dimorphism. This result shows sexual maturation is needed for the tail mechanosensation circuit to acquire its dimorphic properties. We added this to the main text (lines 377-384).

I'd revise the concluding sentence in the mating behavior section (lines 335-337); seeing an effect in an *nmr-1* mutant implies that the mating circuit uses glutamatergic neurotransmission, which is interesting but not terribly scandalous, since many circuits and behaviors use glutamatergic neurotransmission. I'm not sure it could be called a "hijacking" to have a circuit that requires glutamate.

This sentence was removed and was replaced with a different concluding sentence, summarizing additional results that were added (see answers to reviewer #3; lines 374-376). *"These findings demonstrate that nmr-1 and mec-12 work in tandem in males to mediate optimal mating, and suggest that the same mechanisms that mediate tail mechanosensation are also at play in executing appropriate mating behavior."*

In the discussion, the proposal that the sexually dimorphic phenomenon demonstrated in this paper is the result of "extensive sexually dimorphic reorganization during evolution," and "this modality was

important enough for both sexes to withstand the evolutionary pressures that changed its constituents” seems to suggest that there was some ancestral state of a non-dimorphic touch circuit that then changed to the observed situation in *C. elegans*. One of the reasons proposed for this change is “the need to integrate the mating circuit,” which again suggests that there existed an ancestral non-dimorphic state in an animal/nematode that apparently reproduced without mating, and then mating was integrated as an evolutionary novelty. In the absence of any evolutionary evidence in this paper that such an ancestral state exists (ie comparison to male/female species, which are proposed to be the ancestral state, or to the other instances that hermaphroditism has arisen), this seems very speculative. If literature exists suggesting that there is indeed an ancestral non-dimorphic touch circuit in a non-mating species it could be cited to strengthen this argument, otherwise these claims are a bit distracting.

We revised the phrasing without speculating on the evolutionary perspective in this context (lines 412-415): *“It is tempting to speculate that one key driver for the observed dimorphisms in this circuit is the need for males to integrate the mating circuit into the shared nervous system, forcing the redistribution of cells and genes in a different configuration from that of hermaphrodites.”*

Reviewer #2 (Remarks to the Author):

This paper dissects the molecular and cellular architecture of a sexually dimorphic circuit for posterior touch in *C. elegans*. A very interesting aspect of the findings is that males and hermaphrodites use different neural strategies to achieve identical behavioural outputs and that while all circuit components are present in both sexes, each sex relies on different subsets of neurons for touch-dependent behavioural responses. Specifically, the authors use cell-specific silencing of neurons to show that males rely on sensory neurons PHA and PHB and interneuron AVG to elicit forward scribe in response to posterior touch whereas hermaphrodites rely on PHC and PHB sensory neurons and DVA interneuron. Analysis and rescue of mutants show that touch sensation requires the function of *mec-12*, likely in PHA and PHB, and *nmr-1* in AVG in males. In contrast hermaphrodites require the function of *tmc-1*, acting in PHA, and *glr-1*. The role of the identified genes and neurons in male touch sensation is further supported with imaging of neuronal activity in AVG. Finally, the authors propose that a potential reason for the circuit dimorphism may be the necessity of males to integrate other (male-specific) circuits and functions into the tail mechanosensory circuit. This is based on the defective mating behaviour of *nmr-1* mutant males and previous studies which show that AVG interneurons are involved in male mating. Overall, the experiments are convincing, carefully done and support the main conclusions of the paper. In addition, and as

mentioned above, the findings are of great interest not only to those working on sexual dimorphism but also to systems neuroscientists, evolutionary biologists and ethologists.

My main concern is the implication of *mec-12* acting in PHA and PHB in males for behaviour and AVG neural activity. The authors use too broad a promoter (*che-12*) for their *mec-12* rescues. However, they do have PHA and PHB specific promoters, which they use for cell silencing. Could they not use those to confirm that *mec-12* is indeed acting in PHA and PHB? It would be good to see this rescue not just for behaviour but also for neuronal activity in AVG.

We thank the reviewer for his/her suggestions and comments. We addressed the main concern by overexpressing MEC-12 in PHA and PHB separately and specifically, using the *srg-13* and *gpa-6* promoters respectively, and checked the tail-touch responses of *mec-12* mutant males. Interestingly, only PHA re-expression restored the behavior of the males and not PHB (Fig. 2a). We then examined AVG's neuronal activity in those animals and found that the activity is restored compared to *mec-12* mutant males (Supplementary Fig. 9d).

The site of action of *glr-1* is also not reported. Can they rescue the touch response in hermaphrodites by expressing in DVA?

We performed the tail-touch assay on *glr-1* mutant hermaphrodites with overexpression of GLR-1 in DVA specifically. We found that the defective touch responses of the hermaphrodites was not restored as a result of GLR-1 overexpression in DVA (Supplementary Fig. 6). We concluded in the results (line 264) that *glr-1* might be required in a different interneuron.

In addition, to understand the sexually dimorphic circuit architecture it would be informative to see what is the contribution of PHA and PHC to the behaviour observed in AVG sex-reversal experiments. Is the *nmr-1* mutant male enhanced touch response still dependent on PHA? Or is it now dependent on PHC? Can the response of *nmr-1* mutant hermaphrodites be further reduced by silencing PHA? Or does response still rely on PHC?

We thank the reviewer for the suggestion to look deeper into the circuit architecture. We performed PHA or PHC silencing on either hermaphrodites with masculinized AVG or males with feminized AVG in *nmr-1* mutant background. *nmr-1* mutant hermaphrodites with masculinized AVG, which display reduced tail-

touch responses, switched their dependency from PHC to PHA – resembling the function of the male circuit. *nmr-1* mutant males with feminized AVG, which display enhanced tail-touch responses, were not dependent on PHA or PHC (Fig. 4f). This result was added to the main text (lines 248-260).

Minor comment:

In the Discussion, the suggestion of a threshold mechanisms in AVG for behavioural output is not convincing because the results only show that AVG is not required in hermaphrodites for touch response (other neurons may compensate), but AVG may still contribute to behaviour.

This argument was removed from the text.

Reviewer #3 (Remarks to the Author):

Hagar et al report a sexual dimorphism in a mechanosensory circuit of *C.elegans*. As for tail mechanosensation, it is very interesting. But this study still need more clarification before it can be at the level of Nature Communications. More time is required for specific parts to be commented and debated.

Specific Comments:

1. MEC-12 was assayed in the tail mechanosensation of males but no mating defects which may suggest that MEC-12 is not the primary mechanoreceptor as the study is showing with sexual dimorphism.

We thank the reviewer for his/her comments. Throughout the manuscript, we did not suggest that *mec-12* is the primary mechanoreceptor. In fact, *mec-12* is not considered as a mechanoreceptor at all since it is not a membrane receptor, but is an alpha-tubulin protein required for mechanosensation in touch receptor neurons (TRNs) by allowing their 15 protofilament structure (Fukushige et al, 1999). It does not seem to sense the mechanical stimulation in TRNs on its own, but rather enables the appropriate structure for the sensory processing. We strongly believe our results show that MEC-12 fulfils an indispensable role in tail mechanosensation, in a sex and cell-specific manner, at both the behavioral and neuronal levels.

This is supported by several lines of evidence:

1. *mec-12* is needed specifically in males, we now demonstrate that it functions through the tail sensory neuron PHA (Fig. 2, 5, Supplementary Fig. 9d). We speculate on how *mec-12* affects the structure of PHA in the discussion (lines 419-423).

2. Importantly, we added an additional experiment to show the capability of *mec-12* expression to endow sensitivity to tail touch by force-expressing it in PHA in *tmc-1* mutant hermaphrodites, and show it rescues their defective response (Fig. 2b).
3. We now demonstrate that double mutant males for *nmr-1* and *mec-12* are defective also in the time that passed until successful mating (Fig. 7c), revealing a synthetic interaction between the two genes that enhances the mating defects. These findings demonstrate that *nmr-1* and *mec-12* work in tandem in males to mediate optimal mating, and suggest that the same mechanisms that mediate tail mechanosensation are also at play in executing appropriate mating behavior.

Overall, we demonstrate a male-specific role for *mec-12* in tail mechanosensation, and a minor role in mating behavior.

2. The touch assay and response index are not enough to support *tmc-1* tail mechanosensation of hermaphrodite. TMC-1 may regulate resting potential, or form a mechanosensitive complex with other proteins. To clarify this, specific rescue, calcium imaging and ephys recording would be good.

Solid evidence exists to show that homologs of TMC-1 are bona fide mechanosensitive channels (see for example, Pan et al *Neuron* 2018, Jia et al *Neuron* 2020), and proving this physiologically in *C. elegans* is beyond the scope of our manuscript. However, to further support a functional role for *tmc-1* in tail mechanosensation, we now added an experiment to show that force-expressing a *tmc-1* transgene specifically in hermaphrodite PHA (a neuron that normally is not required for tail mechanosensation in hermaphrodites) is sufficient to elicit a tail touch-induced calcium response in the downstream interneuron AVG (again, a neuron that normally is not active in hermaphrodites in this circuit) (Fig. 6a). Furthermore, this transgene was able to partially rescue the tail touch response of *tmc-1* mutant hermaphrodites (Fig. 6b). Overall, we feel that these two results lend considerable new evidence for a functional role for *tmc-1* in tail mechanosensation.

3. Why *tmc-1* required in PHC neurons for hermaphrodite tail mechanosensation but not in PHA neurons? PHA are ciliated neurons but not PHC neurons are different in males (striated rootlets) than hermaphrodite.

We agree that PHA is potentially a good candidate cell to mediate tail touch in hermaphrodites, given that *tmc-1* is known to be expressed in PHA (Taylor et al, 2021) and that the tail-touch responses of *tmc-1*

mutant hermaphrodites could be partially rescued when over-expressing *tmc-1* specifically in PHA (Fig. 6b). However, our cell-specific silencing and rescue experiments point without doubt towards PHC rather than PHA as the key player in our assay. One possible speculation as to why PHC plays a role in tail touch in hermaphrodites but not males could be the recent finding that the PHC dendrite is extended into the tip of the tail in both sexes during juvenile stages, but retracts after sexual maturation only in males (Serrano-Saiz et al, 2017). This could mean that the surface area available for TMC-1 expression throughout the dendrite is higher in hermaphrodite PHC than male PHC.

4.Line 385; “TMC-1 appears to function as a polymodal ion channel, enabling the processing of different types of information”. This is an overstatement in the conclusion as the study failed to show more about *tmc-1*. In addition, TMC-1 may not be a sodium-sensitive channel (L381), according to some published papers.

Thank you for turning our attention to the fact the TMC-1 may not necessarily be a sodium-sensitive channel, we corrected this in the discussion (line 427). As for our argument that TMC-1 might be a polymodal ion channel, we think that since many papers showed TMC-1’s involvement in processing various types of information (elaborated in line 428-432), this is a valid argument. Our paper adds another function for *tmc-1* in *C. elegans*, expanding *tmc-1*’s functional repertoire.

This paragraph has been revised: “*We also found that the ion channel TMC-1 is a hermaphrodite-specific mediator of tail mechanosensation. This finding adds to a growing repertoire of sensory functions attributed to tmc-1 in C. elegans, including salt sensation, avoidance of noxious alkaline environments, egg laying, gentle-nose touch response, and the inhibition of egg-laying in response to a harsh mechanical stimulus*^{31,38,54–56}. Taken together, TMC-1 appears to function as a polymodal ion channel, enabling the processing of different types of information.”

5.Line 178-180; They introduced a sex-determining factor specifically in AVG to switch its sexual identity and connectivity to opposite sex; There no more mention about this “switch” except references.

To elaborate on the sexual identity switch, we added a paragraph explaining the factors in the sex-determination pathway ectopically expressed in AVG for sex reversal (lines 174-183).

6.Harsh touch was not assayed and only optogenetics of AVG were showed.

We assayed harsh tail touch on AVG-silenced animals (males and hermaphrodites) (Fig. 3a) and on AVG-silenced animals in which AVG was sex-reversed (Fig. 3b). These two independent results show that AVG is required for tail mechanosensation only in males, and that this behavior can be shaped by the sexual identity of AVG.

7.AVG evoked Calcium transients in the absence of mechanical stimulus.

In Fig. 5, we demonstrate that posterior mechanical stimulation (“mock”) does not produce a rise in calcium levels in AVG (Fig. 5c). Since AVG does not respond to posterior mechanical stimulation, these recordings represent the calcium transient in a basal state. We think that this is sufficient to show that the fluctuations of calcium levels are very sparse.

REVIEWERS' COMMENTS

Reviewer #1 (Remarks to the Author):

The authors have satisfactorily responded to all of my points, and I have no further concerns or comments regarding the revised version of the manuscript. It represents an exciting addition to the field of sexually dimorphic functions of shared neural circuits.

Reviewer #2 (Remarks to the Author):

The authors have gone to great length to address my initial concerns and those of the other reviewers and the manuscript is much improved. I am satisfied with their data.

Reviewer #3 (Remarks to the Author):

My concerns have been fully addressed, and I think that the revised manuscript fits well to Nature Communications.